# Privately Learning Mixtures of Axis-Aligned Gaussians

**Ishaq Aden-Ali**
Department of Computing and Software
McMaster University
adenali@mcmaster.ca

**Hassan Ashtiani**
Department of Computing and Software
McMaster University
zokaeiam@mcmaster.ca

**Christopher Liaw**
Department of Computer Science
University of Toronto
cvliaw@cs.toronto.edu

## Abstract

We consider the problem of learning mixtures of Gaussians under the constraint of approximate differential privacy. We prove that $\widetilde{O}(k^2 d \log^{3/2}(1/\delta)/\alpha^2\varepsilon)$ samples are sufficient to learn a mixture of $k$ axis-aligned Gaussians in $\mathbb{R}^d$ to within total variation distance $\alpha$ while satisfying $(\varepsilon, \delta)$-differential privacy. This is the first result for privately learning mixtures of unbounded axis-aligned (or even unbounded univariate) Gaussians. If the covariance matrices of each of the Gaussians is the identity matrix, we show that $\widetilde{O}(kd/\alpha^2 + kd \log(1/\delta)/\alpha\varepsilon)$ samples are sufficient.

To prove our results, we design a new technique for privately learning mixture distributions. A class of distributions $\mathcal{F}$ is said to be list-decodable if there is an algorithm that, given "heavily corrupted" samples from $f \in \mathcal{F}$, outputs a list of distributions one of which approximates $f$. We show that if $\mathcal{F}$ is privately list-decodable then we can learn mixtures of distributions in $\mathcal{F}$. Finally, we show axis-aligned Gaussian distributions are privately list-decodable, thereby proving mixtures of such distributions are privately learnable.

## 1 Introduction

The fundamental problem of *distribution learning* concerns the design of algorithms (i.e., *estimators*) that, given samples generated from an unknown distribution $f$, output an "approximation" of $f$. While distribution learning has a long history [54], studying it under privacy constraints is relatively new and unexplored.

In this paper, we work with a rigorous and practical notion of data privacy known as *differential privacy* [36]. Roughly speaking, differential privacy guarantees that no single data point can influence the output of an algorithm too much. Intuitively, this provides privacy by "hiding" the contribution of each individual. Differential privacy is the de facto standard for modern private analysis and has seen widespread impact in both industry and government [12, 22, 28, 29, 39].

In recent years, there has been a flurry of activity in differentially private distribution learning and a number of techniques have been developed in the literature. In the pure differentially private setting, Bun et al. [17] introduced a method to learn classes of distributions that admit a finite cover, i.e. when the class of distributions is well-approximated by a finite number of distributions. They show that this is an exact characterization of distributions which can be learned under pure differential privacy in the sense that a class of distributions is learnable under pure differential privacy if and only if the class

35th Conference on Neural Information Processing Systems (NeurIPS 2021).

admits a finite cover [17, 41]. As a consequence of this result, they obtained pure differentially private algorithms for learning Gaussian distributions provided that the mean of the Gaussians are bounded *and* the covariance matrix of the Gaussians are spectrally bounded.[1] Moreover, such restrictions on the Gaussians are necessary under the constraint of pure differential privacy.

One way to remove the requirement of having a finite cover is to relax to a weaker notion of privacy known as approximate differential privacy. With this notion, Bun et al. [17] introduced a method to learn a class of distributions that, instead of requiring a finite cover, requires a "locally small" cover, i.e. a cover where each distribution in the class is well-approximated by only a small number of elements within the cover. They prove that the class of Gaussians with arbitrary mean and a fixed, known covariance matrix has a locally small cover which implies an approximate differentially private algorithm to learn this class of distributions. Later, Aden-Ali, Ashtiani, and Kamath [4] proved that the class of mean-zero Gaussians (with no assumptions on the covariance matrix) admits a locally small cover leading to an approximate differentially private method to learn the class of all Gaussians.

It is a straightforward observation that if a class of distributions admits a finite cover then the class of its mixtures also admits a finite cover. Combined with the aforementioned work of Bun et al., this implies a pure differentially private algorithm for learning mixtures of Gaussians with bounded mean and spectrally bounded covariance matrices. It is natural to wonder whether an analogous statement holds for locally small covers. In other words, if a class of distributions admits a locally small cover then does the class of mixtures also admit a locally small cover? If so, this would provide a fruitful direction to design differentially private algorithms for learning mixtures of arbitrary Gaussians. Unfortunately, there are simple examples of classes of distributions that admit a locally small cover yet their mixture do *not*. This leaves open the question of designing private algorithms for many classes of distributions that are learnable in the non-private setting. One concrete open problem is for the class of mixtures of two arbitrary univariate Gaussian distributions. A more general problem is private learning of mixtures of $k$ axis-aligned (or general) Gaussian distributions.

## 1.1 Main Results

We demonstrate that it is indeed possible to privately learn mixtures of unbounded univariate Gaussians. More generally, we give sample complexity upper bounds for learning mixtures of unbounded $d$-dimensional axis-aligned Gaussians. In the following theorem and the rest of the paper we use $\widetilde{O}$ to hide polylogarithmic factors, i.e. $\widetilde{O}(f(x))$ means $O(f(x) \log^c f(x))$ for some $c > 0$.

**Theorem 1.1** (Informal). *The sample complexity of learning a mixture of $k$ $d$-dimensional axis-aligned Gaussians to $\alpha$-accuracy in total variation distance under $(\varepsilon, \delta)$-differential privacy is* $\widetilde{O}\left(\frac{k^2 d \log^{3/2}(1/\delta)}{\alpha^2 \varepsilon}\right).$

The formal statement of this theorem can be found in Theorem 5.1. For technical reasons, we do require that $\delta \in (0, 1/n)$ for the above theorem to hold. This condition is quite standard in the differential privacy literature. Indeed, for useful privacy, $\delta$ should be "cryptographically small", i.e., $\delta \ll 1/n$.

Even for the univariate case, our result is the *first* sample complexity upper bound for learning mixture of Gaussians under differential privacy for which the variances are unknown and the parameters of the Gaussians may be unbounded. In the non-private setting, it is known that $\widetilde{\Theta}(kd/\alpha^2)$ samples are necessary and sufficient to learn a mixture of $k$ axis-aligned Gaussian in $\mathbb{R}^d$ [6, 61]. In the private setting, the best known sample complexity lower bound is $\Omega(d/\alpha\varepsilon \log(d))$ under $(\varepsilon, \delta)$-DP when $\delta \leq \widetilde{O}(\sqrt{d}/n)$ [46]. Obtaining improved upper or lower bounds in this setting remains an open question.

If the covariance matrix of each component of the mixture is the same and known or, without loss of generality, equal to the identity matrix, then we can improve the dependence on the parameters and obtain a result that is in line with the non-private setting.

---

[1]When we say that a matrix $\Sigma$ is spectrally bounded, we mean that there are $0 < a_1 \leq a_2$ such that $a_1 \cdot I \preceq \Sigma \preceq a_2 \cdot I$.

**Theorem 1.2** (Informal). *The sample complexity of learning a mixture of $k$ $d$-dimensional Gaussians with identity covariance matrix to $\alpha$-accuracy in total variation distance under $(\varepsilon, \delta)$-differential privacy is $\widetilde{O}\left(\frac{kd}{\alpha^2} + \frac{kd \log(1/\delta)}{\alpha \varepsilon}\right)$.*

We relegate the formal statement and the proof of this theorem to the supplementary materials (see Appendix F). Note that the work of [53] implies an upper bound of $O(k^2 d^3 \log^2(1/\delta)/\alpha^2 \varepsilon^2)$ for private learning of the same class albeit in the incomparable setting of parameter estimation.

**Comparison with locally small covers.** While the results in [4, 17] for learning Gaussian distributions under approximate differential privacy do not yield finite-time algorithms, they do give strong information-theoretic upper bounds. This is achieved by showing that certain classes of Gaussians admit locally small covers. It is thus natural to ask if we can obtain sharper results by showing that mixtures of Gaussians also admit locally small covers. Unfortunately, the following simple example shows that not even mixtures of two univariate Gaussians admit locally small covers.

**Proposition 1.3** (Informal version of Proposition B.6). *Every cover for the class of mixtures of two univariate Gaussians is not locally small.*

## 1.2 Techniques

To prove our result, we devise a novel technique which reduces the problem of privately learning mixture distributions to the problem of private list-decodable learning of distributions. The framework of list-decodable learning was introduced by Balcan, Blum, and Vempala [8] and Balcan, Röglin, and Teng [9] in the context of clustering but has since been studied extensively in the literature in a number of different contexts [7, 20, 21, 26, 27, 49, 55, 56]. The problem of list-decodable learning of distributions is as follows. There is a distribution $f$ of interest that we are aiming to learn. However, we do not receive samples from $f$; rather we receive samples from a *corrupted* distribution $g = (1 - \gamma)f + \gamma h$ where $\gamma \in (0, 1)$ and $h$ is some arbitrary distribution. In our application, $\gamma$ is close to 1, i.e. *most* of the samples are corrupted. The goal in list-decodable learning is to output a *short* list of distributions $f_1, \ldots, f_m$ with the requirement that $f$ is close to at least one of the $f_i$'s. The formal definition of list-decodable learning can be found in Definition 2.6. Informally, the reduction can be summarized by the following theorem which is formalized in Section 3.

**Theorem 1.4** (Informal). *If a class of distributions $\mathcal{F}$ is privately list-decodable then mixtures of distributions from $\mathcal{F}$ are privately learnable.*

Roughly speaking, the reduction from learning mixtures of distribution to list-decodable learning works as follows. Suppose that there is an unknown distribution $f$ which is a mixture of $k$ distributions $f_1, \ldots, f_k$. A list-decodable learner would then receive samples from $f$ as input and output a short list of distributions $\widehat{\mathcal{F}}$ so that for every $f_i$ there is some element in $\widehat{\mathcal{F}}$ that is close to $f_i$. In particular, some mixture of distributions from $\widehat{\mathcal{F}}$ must be close to the true distribution $f$. Since $\widehat{\mathcal{F}}$ is a small finite set, the set of possible mixtures must also be relatively small. This last observation allows us to make use of private hypothesis selection which selects a good hypothesis from a small set of candidate hypotheses [4, 17]. In Section 3, we describe the aforementioned reduction in more detail. We note that a similar connection between list-decodable learning and learning mixture distributions was also used by Diakonikolas et al. [26]. However, our reduction is focused on the private setting.

The reduction shows that to privately learn mixtures, it is sufficient to design differentially private list-decodable learning algorithms that work for (corrupted versions of) the individual mixture components. To devise list-decodable learners for (corrupted) univariate Gaussian, we utilize "stability-based" histograms [15, 51] that satisfy approximate differential privacy.

To design a list-decodable learner for corrupted univariate Gaussians, we follow a three-step approach that is inspired by the seminal work of Karwa and Vadhan [50]. First, we use a histogram to output a list of variances one of which approximates the true variance of the Gaussian. As a second step, we would like to output a list of means which approximate the true mean of the Gaussian. This can be done using histograms provided that we roughly know the variance of the Gaussian. Since we have candidate variances from the first step, we can use a sequence of histograms where the width of the bins of each of the histograms is determined by the candidate variances from the first step. As a last step, using the candidate variances and means from the first two steps, we are able to construct a small set of distributions one of which approximates the true Gaussian to within accuracy $\alpha$. In the

axis-aligned Gaussians setting, we use our solution for the univariate case as a subroutine on each dimension separately. Now that we have a list-decodable learner for axis-aligned Gaussians, we use our reduction to obtain a private learning algorithm for learning mixtures of axis-aligned Gaussians.

Approaches based on constructing lists of candidate variances and means in order to learn an accurate mixture have been previously considered in the non-private setting [5, 6, 23, 61]. However, it does not seem possible to directly privatize the algorithms in this line of work since they construct these candidates directly from the samples, which is a clear violation of privacy.

### 1.3 Open Problems

The most basic open problem is to understand the exact sample complexity (up to constants) for learning mixtures of univariate Gaussians under approximate differential privacy.

**Conjecture 1.5** (Informal). *The sample complexity of learning a mixture of $k$ univariate Gaussians to within total variation distance $\alpha$ with high probability under $(\varepsilon, \delta)$-DP is $\Theta\left(\frac{k}{\alpha^2} + \frac{k}{\alpha\varepsilon} + \frac{\log(1/\delta)}{\varepsilon}\right)$.*

Another open question is whether it is possible to privately learn mixtures of (arbitrary) high-dimensional Gaussians. We conjecture that it is possible and with the following sample complexity.

**Conjecture 1.6** (Informal). *The sample complexity of learning a mixture of $k$ $d$-dimensional Gaussians to within total variation distance $\alpha$ with high probability under $(\varepsilon, \delta)$-DP is $\Theta\left(\frac{kd^2}{\alpha^2} + \frac{kd^2}{\alpha\varepsilon} + \frac{\log(1/\delta)}{\varepsilon}\right)$.*

### 1.4 Additional Related Work

Recently, [17] showed how to learn spherical Gaussian mixtures where each Gaussian component has bounded mean under pure differential privacy. Acharya, Sun and Zhang [3] were able to obtain lower bounds in the same setting that nearly match the upper bounds of Bun et al. [17]. Both [47, 53] consider differentially private learning of Gaussian mixtures, however their focus is on parameter estimation and therefore require additional assumptions such as separation or boundedness of the components.

There has been a flurry of activity on differentially private distribution learning and parameter estimation in recent years for many problem settings [3, 11, 14, 16–18, 25, 30, 38, 46, 48, 50, 52, 53, 59, 60]. There has also been a lot of work in the locally private setting [2, 31–33, 40, 43, 44, 63, 64]. Other work on differentially private estimation include [1, 10, 13, 19, 34, 58, 65]. For a more comprehensive review of differentially private statistics, see [45].

## 2 Preliminaries

For any $m \in \mathbb{N}$, $[m]$ denotes the set $\{1, 2, \ldots, m\}$. Let $X \sim f$ denote a random variable $X$ sampled from the distribution $f$. Let $(X^i)_{i=1}^m \sim f^m$ denote an i.i.d. random sample of size $m$ from distribution $f$. For a vector $x \in \mathbb{R}^d$, we refer to the $i$th element of vector $x$ as $x_i$. For any $k \in \mathbb{N}$, we define the $k$-dimensional probability simplex to be $\Delta_k := \{(w_1, \ldots, w_k) \in \mathbb{R}_{\geq 0}^k : \sum_{i=1}^k w_i = 1\}$. For a vector $\mu \in \mathbb{R}^d$ and a positive semidefinite matrix $\Sigma$, we use $\mathcal{N}(\mu, \Sigma)$ to denote the multivariate normal distribution with mean $\mu$ and covariance matrix $\Sigma$. We define $\mathcal{G}$ to be the class of univariate Gaussians and $\mathcal{G}^d = \{\mathcal{N}(\mu, \Sigma) : \Sigma_{ij} = 0 \ \forall i \neq j \text{ and } \Sigma_{ii} > 0 \ \forall i\}$ to be the class of axis-aligned Gaussians.

**Definition 2.1** ($k$-mix$(\mathcal{F})$). *Let $\mathcal{F}$ be a class of probability distributions. Then the class of $k$-mixtures of $\mathcal{F}$, written $k$-mix$(\mathcal{F})$, is defined as*

$$k\text{-mix}(\mathcal{F}) := \left\{ \sum_{i=1}^k w_i f_i : (w_1, \ldots, w_k) \in \Delta_k, f_1, \ldots, f_k \in \mathcal{F} \right\}.$$

### 2.1 Distribution Learning

A *distribution learning method* is a (potentially randomized) algorithm that, given a sequence of i.i.d. samples from a distribution $f$, outputs a distribution $\widehat{f}$ as an estimate of $f$. The focus of this paper is on absolutely continuous probability distributions (distributions that have a density with respect to

the Lebesgue measure), so we refer to a probability distribution and its probability density function interchangeably. The specific measure of "closeness" between distributions that we use is the *total variation (TV) distance*.

**Definition 2.2.** *Let $g$ and $f$ be two probability distributions defined over $\mathcal{X}$ and let $\Omega$ be the Borel $\sigma$-algebra on $\mathcal{X}$. The* total variation distance *between $g$ and $f$ is defined as*

$$d_{\mathrm{TV}}(g, f) = \sup_{S \in \Omega} |\mathbf{P}_g(S) - \mathbf{P}_f(S)| = \frac{1}{2} \int_{x \in \mathcal{X}} |g(x) - f(x)| \mathrm{d}x = \frac{1}{2} \|g - f\|_1 \in [0, 1].$$

*where $\mathbf{P}_g(S)$ denotes the probability measure that $g$ assigns to $S$. Moreover, if $\mathcal{F}$ is a set of distributions over a common domain, we define $d_{\mathrm{TV}}(g, \mathcal{F}) = \inf_{f \in \mathcal{F}} d_{\mathrm{TV}}(g, f)$.*

**Definition 2.3** (PAC learner)**.** *We say algorithm $\mathcal{A}$ is a* PAC-learner *for a class of distributions $\mathcal{F}$ which uses $m(\alpha, \beta)$ samples, if for every $\alpha, \beta \in (0, 1)$, every $f \in \mathcal{F}$, and every $n \geq m(\alpha, \beta)$ the following holds: if the algorithm is given parameters $\alpha, \beta$ and a sequence of $n$ i.i.d. samples from $f$ as inputs, then it outputs an approximation $\widehat{f}$ such that $d_{\mathrm{TV}}(f, \widehat{f}) \leq \alpha$ with probability at least $1 - \beta$.[2]*

We work with an additive corruption model often studied in the list-decodable setting that is inspired by the seminal work of Huber [42]. In this model, a sample is drawn from a distribution of interest with some probability, and with the remaining probability is drawn from an arbitrary distribution. Our list-decodable learners take samples from these "corrupted" distributions as input.

**Definition 2.4** ($\gamma$-corrupted distributions)**.** *Fix some distribution $f$ and let $\gamma \in (0, 1)$. We define a $\gamma$-corrupted distribution of $f$ as any distribution $g$ such that $g = (1 - \gamma)f + \gamma h$ for an arbitrary distribution $h$. We define $\mathcal{H}_\gamma(f)$ to be the set of all $\gamma$-corrupted distributions of $f$.*

**Remark 2.5.** *Observe that $\mathcal{H}_\gamma(f)$ is monotone increasing in $\gamma$, i.e. $\mathcal{H}_\gamma(f) \subset \mathcal{H}_{\gamma'}(f)$ for all $\gamma' \in (\gamma, 1)$. To see this, note that if $g = (1 - \gamma)f + \gamma h$ then we can also rewrite $g = (1 - \gamma')f + \gamma' h'$, where $h' = \frac{\gamma' - \gamma}{\gamma} f + \frac{\gamma}{\gamma'} h$. Hence, $g \in C_{\gamma'}(f)$.*

In this work, $\gamma$ is usually quite close to 1, i.e. the vast majority of the samples are corrupted. Next, we define list-decodable learning. In this setting, the goal is to learn a distribution $f$ given samples from a $\gamma$-corrupted distribution $g$ of $f$. As $\gamma \approx 1$, the goal is to output a list of distributions, one of which approximates $f$. We use this primitive to design algorithms for learning mixture distributions.

**Definition 2.6** (list-decodable learner)**.** *We say algorithm $\mathcal{A}_{\mathrm{LIST}}$ is an $L$-list-decodable learner for a class of distributions $\mathcal{F}$ using $m_{\mathrm{LIST}}(\alpha, \beta, \gamma)$ samples if for every $\alpha, \beta, \gamma \in (0, 1)$, $n \geq m_{\mathrm{List}}(\alpha, \beta, \gamma)$, $f \in \mathcal{F}$, and $g \in \mathcal{H}_\gamma(f)$, the following holds: given parameters $\alpha, \beta, \gamma$ and a sequence of $n$ i.i.d. samples from $g$ as inputs, $\mathcal{A}_{\mathrm{LIST}}$ outputs a set of distributions $\widetilde{\mathcal{F}}$ with $|\widetilde{\mathcal{F}}| \leq L$ such that with probability no less than $1 - \beta$ we have $d_{\mathrm{TV}}(f, \widetilde{\mathcal{F}}) \leq \alpha$.*

## 2.2 Differential Privacy

Let $X^* = \cup_{i=1}^{\infty} X^i$ be the set of all datasets of arbitrary size over a domain set $X$. Two datasets $D, D' \in X^*$ are *neighbours* if $D$ and $D'$ differ in at most one data point. Informally, an algorithm is differentially private if its output on neighbouring databases are similar. Formally, differential privacy (DP)[3] has the following definition.

**Definition 2.7** ([35, 36])**.** *A randomized algorithm $T : X^* \to \mathcal{Y}$ is $(\varepsilon, \delta)$-differentially private if for all $n \geq 1$, for all neighbouring datasets $D, D' \in X^n$, and for all measurable subsets $S \subseteq \mathcal{Y}$,*

$$\Pr\left[T(D) \in S\right] \leq e^\varepsilon \Pr[T(D') \in S] + \delta.$$

*If $\delta = 0$, we say that $T$ is $\varepsilon$-differentially private.*

We refer to $\varepsilon$-DP as *pure* DP, and $(\varepsilon, \delta)$-DP for $\delta > 0$ as *approximate* DP. We make use of the following property of differentially private algorithms which asserts that adaptively composing differentially private algorithms remains differentially private. By adaptive composition, we mean that we run a sequence of algorithms $M_1(D), \dots, M_T(D)$ where the choice of algorithm $M_t$ may depend on the outputs of $M_1(D), \dots, M_{t-1}(D)$.

---

[2]The probability is over $m(\alpha, \beta)$ samples drawn from $f$ and the randomness of the algorithm.

[3]We will use the acronym DP to refer to both the terms "differential privacy" and "differentially private". Which term we are using will be clear from the specific sentence.

**Lemma 2.8** (Composition of DP [36, 37]). *If $M$ is an adaptive composition of differentially private algorithms $M_1, \ldots, M_T$ then the following two statements hold:*

*1. If $M_1, \ldots, M_T$ are $(\varepsilon_1, \delta_1), \ldots, (\varepsilon_T, \delta_T)$-DP, then $M$ is $(\varepsilon, \delta)$-DP for*

$$\varepsilon = \sum_{t=1}^{T} \varepsilon_t \quad and \quad \delta = \sum_{t=1}^{T} \delta_t.$$

*2. If $M_1, \ldots, M_T$ are $(\varepsilon_0, \delta_1), \ldots, (\varepsilon_0, \delta_T)$-DP for some $\varepsilon_0 \leq 1$, then for any $\delta_0 > 0$, $M$ is $(\varepsilon, \delta)$-DP for*

$$\varepsilon = \varepsilon_0 \sqrt{6T \log(1/\delta_0)} \quad and \quad \delta = \delta_0 + \sum_{t=1}^{T} \delta_t.$$

The first statement in Lemma 2.8 is often referred to as *basic* composition and the second statement is often referred to as *advanced* composition. We also make use of the fact that post-processing the output of a differentially private algorithm does not impact privacy.

**Lemma 2.9** (Post Processing). *If $M : \mathcal{X}^n \to \mathcal{Y}$ is $(\varepsilon, \delta)$-differentially private, and $P : \mathcal{Y} \to \mathcal{Z}$ is any randomized function, then the algorithm $P \circ M$ is $(\varepsilon, \delta)$-differentially private.*

We define $(\varepsilon, \delta)$-DP PAC learners and $(\varepsilon, \delta)$-DP $L$-list-decodable learners as PAC learners and $L$-list-decodable learners that satisfy $(\varepsilon, \delta)$-DP.

## 3 List-decodability and Learning Mixtures

In this section, we describe our general technique which reduces the problem of private learning of mixture distributions to private list-decodable learning of distributions. We show that if we have a differentially private list-decodable learner for a class of distributions then this can be transformed, in a black-box way, to a differentially private PAC learner for the class of *mixtures* of such distributions. In the next section, we describe private list-decodable learners for the class of Gaussians and thereby obtain private algorithms for learning mixtures of Gaussians.

First, let us begin with some intuition in the *non*-private setting. Suppose that we have a distribution $g$ which can be written as $g = \sum_{i=1}^{k} \frac{1}{k} f_i$. Then we can view $g$ as a $\frac{k-1}{k}$-corrupted distribution of $f_i$ for each $i \in [k]$. Any list-decodable algorithm that receives samples from $g$ as input is very likely to output a candidate set $\widehat{\mathcal{F}}$ which contains distributions that are close to $f_i$ for each $i \in [k]$. Hence, if we let $\mathcal{K} = \{\sum_{i \in [k]} \frac{1}{k} \widehat{f_i} : \widehat{f_i} \in \widehat{\mathcal{F}}\}$, then $g$ must be close to some distribution in $\mathcal{K}$. The only remaining task is to find a distribution in $\mathcal{K}$ that is close to $g$; this final task is known as hypothesis selection and has a known solution [24]. We note that the above argument can be easily generalized to the setting where $g$ is a non-uniform mixture, i.e. $g = \sum_{i=1}^{k} w_i f_i$ where $(w_1, \ldots, w_k) \in \Delta_k$.

The above establishes a blueprint that we can follow in order to obtain a private learner for mixture distributions. In particular, we aim to come up with a private list-decoding algorithm which receives samples from $f$ to produce a set $\widehat{\mathcal{F}}$. Thereafter, we can construct a candidate set $\mathcal{K}$ as mixtures of distributions from $\widehat{\mathcal{F}}$. Note that this step does not access the samples and therefore maintains privacy. In order to choose a good candidate from $\mathcal{K}$, we make use of private hypothesis selection [4, 17].

Formally, the following theorem establishes the reduction from private list-decodable learning to learning of mixtures. The proof can be found in Appendix C of the supplementary materials.

**Theorem 3.1.** *Let $k \in \mathbb{N}$ and $\varepsilon, \delta \in (0, 1)$. If $\mathcal{F}$ is $(\varepsilon/2, \delta)$-DP $L$-list-decodable with $m_{\text{LIST}}$ samples then there is an $(\varepsilon, \delta)$-DP PAC learner for $k\text{-mix}(\mathcal{F})$ where the number of samples used is $m(\alpha, \beta, \varepsilon, \delta) =$*

$$m_{\text{LIST}}\left(\frac{\alpha}{18}, \frac{\beta}{2k}, 1 - \frac{\alpha}{18k}, \frac{\varepsilon}{2}, \delta\right) + O\left(\frac{k \log(Lk/\alpha) + \log(1/\beta)}{\alpha^2} + \frac{k \log(Lk/\alpha) + \log(1/\beta)}{\alpha \varepsilon}\right).$$

This reduction is quite useful because it is conceptually much simpler to devise list-decodable learners for a given class $\mathcal{F}$. In what follows, we will devise such list-decodable learners for certain classes and use Theorem 3.1 to obtain private PAC learners for mixtures of these classes.

## 4 Learning Univariate Gaussian Mixtures

Let $\mathcal{G}$ be the class of all univariate Gaussians. In this section we consider the problem of privately learning univariate Guassian Mixtures, $k\text{-mix}(\mathcal{G})$. In the previous section, we showed that it is

sufficient to design private list-decodable learners for univariate Gaussians. As a warm-up and to build intuition about our techniques, we begin with the simpler problem of constructing private list-decodable learners for Gaussians with a single known variance $\sigma^2$. In what follows, we often use "tilde" (e.g. $\widetilde{M}, \widetilde{V}$) to denote sets that are meant to be *coarse*, or *constant*, approximations and "hat" (e.g. $\widehat{\mathcal{F}}, \widehat{M}, \widehat{V}$) to denote sets that are meant to be *fine*, say $O(\alpha)$, approximations.

## 4.1 Warm-up: Learning Gaussian Mixtures with a Known, Shared Variance

In this sub-section we construct a private list-decodable learner for univariate Gaussians with a known variance $\sigma^2$. A useful algorithmic primitive that we will use throughout this section and the next is the *stable histogram* algorithm.

**Lemma 4.1** (Histogram learner [15, 51])**.** *Let $n \in \mathbb{N}$, $\eta, \beta, \varepsilon \in (0,1)$ and $\delta \in (0, 1/n)$. Let $D$ be a dataset of $n$ points over a domain $\mathcal{X}$. Let $K$ be a countable index set and $\mathbf{B} = \{B_i\}_{i \in K}$ be a collection of disjoint bins defined on $\mathcal{X}$, i.e. $B_i \subseteq \mathcal{X}$ and $B_i \cap B_j = \emptyset$ for $i \neq j$. Finally, let $\overline{p}_i = \frac{1}{n} \cdot |D \cap B_i|$. There is an $(\varepsilon, \delta)$-DP algorithm* `Stable-Histogram`$(\varepsilon, \delta, \eta, \beta, D, \mathbf{B})$ *that takes as input parameters $\varepsilon, \delta, \eta, \beta$, dataset $D$ and bins $\mathbf{B}$, and outputs estimates $\{\widetilde{p}_i\}_{i \in K}$ such that $|\overline{p}_i - \widetilde{p}_i| \leq \eta$ for all $i \in K$ with probability no less than $1 - \beta$ so long as $n = \Omega\left(\frac{\log(1/\beta\delta)}{\eta\varepsilon}\right)$.*

For any fixed $\sigma^2 > 0$ we define $\mathcal{G}_\sigma$ to be the set of all univariate Gaussians with variance $\sigma^2$. For the remainder of this section, we let $g = \mathcal{N}(\mu, \sigma^2) \in \mathcal{G}_\sigma$ and $g' \in \mathcal{H}_\gamma(g)$. (Recall that $g' \in \mathcal{H}_\gamma(g)$ means that $g' = (1 - \gamma)g + \gamma h$ for some distribution $h$.) Algorithm 1 shows how we privately output a list of real numbers, one of which is close to the mean of $g$ given samples from $g'$. The following lemma shows that the output of Algorithm 1 is a list of real numbers with the guarantee that at least one element in the list is close to the true mean of a Gaussian which has been corrupted. Note that the lemma assumes the slightly weaker condition where the algorithm receives an approximation to the standard deviation instead of the true standard deviation. This additional generality is used in the next section.

**Lemma 4.2.** *Algorithm 1 is an $(\varepsilon, \delta)$-DP algorithm such that for any $g = \mathcal{N}(\mu, \sigma^2)$ and $g' \in \mathcal{H}_\gamma(g)$, when it is given parameters $\varepsilon, \beta, \gamma \in (0, 1)$, $\delta \in (0, 1/n)$, $\widetilde{\sigma} \in [\sigma, 2\sigma)$ and dataset $D$ of $n$ i.i.d. samples from $g'$ as input, it outputs a set $\widetilde{M}$ of real numbers of size $|\widetilde{M}| \leq \frac{12}{1-\gamma}$. Furthermore, with probability $1 - \beta$ there is an element $\widetilde{\mu} \in \widetilde{M}$ such that $|\widetilde{\mu} - \mu| \leq \sigma$, so long as $n = \Omega\left(\frac{\log(1/\beta\delta)}{(1-\gamma)\varepsilon}\right)$.*

---

**Algorithm 1:** `Univariate-Mean-Decoder`$(\beta, \gamma, \varepsilon, \delta, \widetilde{\sigma}, D)$.

**Input** : Parameters $\varepsilon, \beta, \gamma \in (0, 1)$, $\delta \in (0, 1/n)$, $\widetilde{\sigma}$ and dataset $D$

**Output** : Set of approximate means $\widetilde{M}$.

1 Partition $\mathbb{R}$ into bins $\mathbf{B} = \{B_i\}_{i \in \mathbb{N}}$ where $B_i = ((i - 0.5)\widetilde{\sigma}, (i + 0.5)\widetilde{\sigma}]$.

2 $\{\widetilde{p}_i\}_{i \in \mathbb{N}} \leftarrow$ `Stable-Histogram`$(\varepsilon, \delta, (1 - \gamma)/24, \beta/2, D, \mathbf{B})$.

3 $H \leftarrow \{i : \widetilde{p}_i > (1 - \gamma)/8\}$

4 If $|H| > 12/(1 - \gamma)$ **fail** and return $\widetilde{M} = \emptyset$

5 $\widetilde{M} \leftarrow \{i\widetilde{\sigma} : i \in H\}$

6 **Return** $\widetilde{M}$.

---

We begin by gathering several simple claims whose proofs can be found in Appendix D.1. Let $p_i = \mathbf{P}_{X \sim g'}[X \in B_i]$ be the probability that a sample drawn from $g'$ lands in bin $B_i$. Let $\overline{p}_i = \frac{1}{n}|D \cap B_i|$ be the actual number of samples drawn from $g'$ that have landed in $B_i$. Let $j = \lceil \mu/\widetilde{\sigma} \rceil$. It is a simple calculation to check that $|j\widetilde{\sigma} - \mu| \leq \sigma$. Thus, we would like to show that $j\widetilde{\sigma} \in \widetilde{M}$ or, equivalently, that $j \in H$. A straightforward calculation shows that $p_j \geq (1 - \gamma)/3$. Then a standard application of a Chernoff bound shows that many samples actually land in bin $B_j$, as asserted by the following claim.

**Claim 4.3.** *If $n = \Omega(\log(1/\beta)/(1 - \gamma))$ then $\overline{p}_j > (1 - \gamma)/6$ with probability at least $1 - \beta/2$.*

Next, we claim that the output of the stable histogram approximately preserves the weight of all the bins and, moreover, that the output does not have too many heavy bins. The first assertion implies

that since bin $B_j$ is heavy, the stable histogram also determines that bin $B_j$ is heavy. The second assertion implies that the algorithm does not fail. Let $\{\widetilde{p}_i\}_{i \in \mathbb{N}}$ be the output of the stable histogram, as defined in Algorithm 1.

**Claim 4.4.** *If $n = \Omega(\log(1/\beta\delta)/(1-\gamma)\varepsilon)$ then with probability $1 - \beta/2$, we have (i) $|\overline{p}_i - \widetilde{p}_i| \le (1-\gamma)/24$ for all $i \in \mathbb{N}$ and (ii) $|H| = |\{i \in \mathbb{N} : \widetilde{p}_i > (1-\gamma)/8\}| \le 12/(1-\gamma)$.*

With Claim 4.3 and Claim 4.4 in hand, we are now ready to prove Lemma 4.2.

*Proof of Lemma 4.2.* We briefly prove that the algorithm is private before proceeding to the other assertions of the lemma.

**Privacy.** Line 2 is the only part of the algorithm that looks at the data and it is $(\varepsilon, \delta)$-DP by Lemma 4.1. The remainder of the algorithm can be viewed as post-processing (Lemma 2.9) so it does not affect the privacy.

**Bound on $|\widetilde{M}|$.** For the bound on $|\widetilde{M}|$, observe that if $|H| > 12/(1-\gamma)$ then the algorithm fails so $|\widetilde{M}| \le 12/(1-\gamma)$ deterministically.

**Utility.** Let $g, g', \mu$ be as defined in the statement of the lemma. We now show that there exists $\widetilde{\mu} \in \widetilde{M}$ such that $|\widetilde{\mu} - \mu| \le \sigma$. Let $j = \lceil \mu/\widetilde{\sigma} \rceil$. For the remainder of the proof, we assume that $n = \Omega(\log(1/\beta\delta)/(1-\gamma)\varepsilon)$.

Claim 4.3 asserts that, with probability $1 - \beta/2$, we have $\overline{p}_j > (1-\gamma)/6$. Claim 4.4 asserts that, with probability $1 - \beta/2$, $\widetilde{p}_j \ge \overline{p}_j - (1-\gamma)/24$ *and that* $|H| \le 12/(1-\gamma)$. By a union bound, with probability $1 - \beta$, we have that $\overline{p}_j > (1-\gamma)/8$ and the algorithm does not fail. This implies that $j \in H$ so $j\widetilde{\sigma} \in \widetilde{M}$. Finally, note that $|j\widetilde{\sigma} - \mu| \le \widetilde{\sigma}/2 \le \sigma$ where the last inequality uses the assumption that $\widetilde{\sigma} \le 2\sigma$. $\qquad\square$

We can now use Lemma 4.2 to get a private list-decodable learner (Corollary 4.5) and then use this private list decodable learner together with our reduction (Theorem 3.1) to get an $(\varepsilon, \delta)$-PAC learner for $k$-mix$(\mathcal{G})$ (Theorem 4.6). The proof of Corollary 4.5 can be found in Appendix D.

**Corollary 4.5.** *For any $\varepsilon \in (0, 1)$ and $\delta \in (0, 1/n)$, there is an $(\varepsilon, \delta)$-DP $L$-list-decodable learner for $\mathcal{G}_\sigma$ with known $\sigma > 0$ where $L = O\left(\frac{1}{(1-\gamma)\alpha}\right)$, and the number of samples used is $m_{\mathrm{LIST}}(\alpha, \beta, \gamma, \varepsilon, \delta) = O\left(\frac{\log(1/\beta\delta)}{(1-\gamma)\varepsilon}\right)$.*

**Theorem 4.6.** *For any $\varepsilon \in (0, 1)$ and $\delta \in (0, 1/n)$, there is an $(\varepsilon, \delta)$-DP PAC learner for $k$-mix$(\mathcal{G}_\sigma)$ with known $\sigma > 0$ that uses $m(\alpha, \beta, \varepsilon, \delta) = \widetilde{O}\left(\frac{k + \log(1/\beta)}{\alpha^2} + \frac{k\log(1/\beta\delta)}{\alpha\varepsilon}\right)$ samples.*

## 4.2 Learning Arbitrary Univariate Gaussian Mixtures

In this subsection, we construct a list-decodable learner for $\mathcal{G}$, the class of all univariate Gaussians. First, in Lemma 4.7, we design an $(\varepsilon, \delta)$-DP algorithm that receives samples from $g' \in \mathcal{H}_\gamma(g)$ where $g \in \mathcal{G}$ and outputs a list of candidate values for the standard deviation, one of which approximates the standard deviation of $g$ with high probability. Then, in Lemma 4.8, we use Lemma 4.2 and Lemma 4.7 to design an $(\varepsilon, \delta)$-DP list-decoder for $\mathcal{G}$.

### 4.2.1 Estimating the variance

We begin with a method to estimate the variance. Here, we provide a high-level overview and relegate the details to the appendix. Suppose that $g = \mathcal{N}(\mu, \sigma^2)$ and $g' \in \mathcal{H}_\gamma(g)$. Our goal is to obtain a *multiplicative* estimate of $\sigma^2$. The key observation is as follows. If $X_1, X_2 \sim g'$ then $Y = (X_1 - X_2)/\sqrt{2}$ is distributed as $(1-\gamma)^2\mathcal{N}(0, \sigma^2) + (1 - (1-\gamma)^2)h$ for some distribution $h$. In particular, with probability roughly $(1-\gamma)^2$, $|Y|$ is itself a good estimate of $\sigma$. This observation allows us to proceed similarly to the proof of Lemma 4.2 with two changes. First, since our goal is a *multiplicative* approximation to $\sigma^2$, we use bins of the form $(2^i, 2^{i+1}]$ for $i \in \mathbb{N}$. Second, given a dataset $D = \{X_1, \ldots, X_{2m}\}$, we transform it to the dataset $D' = \{|X_1 - X_2|/\sqrt{2}, \ldots, |X_{2m-1} - X_{2m}|/\sqrt{2}\}$ and use the previously mentioned observation. The following lemma formalizes the guarantee that we can achieve. The proof can be found in Appendix D.2.

**Lemma 4.7.** *There is an $(\varepsilon, \delta)$-DP algorithm such that for any $g = \mathcal{N}(\mu, \sigma^2)$ and $g' \in \mathcal{H}_\gamma(g)$, when it is given parameters $\varepsilon, \beta, \gamma \in (0, 1)$, $\delta \in (0, 1/n)$ and dataset $D$ of $2n$ i.i.d. samples from $g'$ as input, it outputs a set $\widetilde{V}$ of positive real numbers of size $|\widetilde{V}| \leq \frac{12}{(1-\gamma)^2}$. Furthermore, with probability no less than $1 - \beta$ there is an element $\widetilde{\sigma} \in \widetilde{V}$ such that $\sigma \leq \widetilde{\sigma} < 2\sigma$, so long as $n = \Omega\left(\frac{\log(1/\beta\delta)}{(1-\gamma)^2\varepsilon}\right)$.*

#### 4.2.2 A list-decodable learner for univariate Gaussians

We can now use Lemma 4.2 and Lemma 4.7 to design a list-decodable learner for $\mathcal{G}$. This is done in a few steps. First, we obtain a list of candidate variances using Lemma 4.7. We know that one of these candidates is a good approximation to the true variance although we may not know which one. As a second step, we use the algorithm implied by Lemma 4.2 for *all* the candidate variances to get a list of candidate means. Since one of the candidate variances is a good estimate of the true variance, Lemma 4.2 promises that one of the candidate means is a good estimate of the true mean. Given a list of candidate variances and candidate means, we can create a list of all pairs of variances and means to obtain a list of distributions such that one is close to the true Gaussian. The guarantee is formalized in the following lemma and the proof appears in Appendix D.3.

**Lemma 4.8.** *There is an $(\varepsilon, \delta)$-DP algorithm such for any $g = \mathcal{N}(\mu, \sigma^2)$ and $g' \in \mathcal{H}_\gamma(g)$, when it is given parameters $\varepsilon, \alpha, \beta, \gamma \in (0, 1)$, $\delta \in (0, 1/n)$ and dataset $D$ of $n$ i.i.d. samples from $g'$ as inputs, it outputs a set $\widehat{M}$ of real numbers and a set $\widehat{V}$ of positive real numbers such that*

$$|\widehat{M}| \leq \frac{144 \cdot (2 \cdot \lceil 1/\alpha \rceil + 1)}{(1-\gamma)^3} \quad and \quad |\widehat{V}| \leq \frac{12 \cdot \lceil \log_{1+\alpha}(2) \rceil}{(1-\gamma)^2}.$$

*Furthermore, with probability no less than $1 - \beta$, we have the following:*

1. *$\exists \widehat{\mu} \in \widehat{M}$ such that $|\widehat{\mu} - \mu| \leq \alpha\sigma$*
2. *$\exists \widehat{\sigma} \in \widehat{V}$ such that $|\widehat{\sigma} - \sigma| \leq \alpha\sigma$*

*so long as $n = \widetilde{\Omega}\left(\frac{\log^{3/2}(1/\beta\delta)}{(1-\gamma)^2\varepsilon}\right)$.*

We can now use Lemma 4.8 to get the following result. We defer the proof to Appendix D.4.

**Corollary 4.9.** *For any $\varepsilon \in (0, 1)$ and $\delta \in (0, 1/n)$, there is an $(\varepsilon, \delta)$-DP $L$-list-decodable learner for $\mathcal{G}$ where $L = O\left(\frac{1}{(1-\gamma)^5\alpha^2}\right)$ and the number of samples used by the algorithm is $m_{\mathrm{LIST}}(\alpha, \beta, \gamma, \varepsilon, \delta) = \widetilde{O}\left(\frac{\log^{3/2}(1/\beta\delta)}{(1-\gamma)^2\varepsilon}\right)$.*

Finally, Corollary 4.9 and Theorem 3.1 immediately imply the following theorem.

**Theorem 4.10.** *For any $\varepsilon \in (0, 1)$ and $\delta \in (0, 1/n)$, there is an $(\varepsilon, \delta)$-DP PAC learner for $k$-mix$(\mathcal{G})$ that uses $m(\alpha, \beta, \varepsilon, \delta) = \widetilde{O}\left(\frac{k^2 \log^{3/2}(1/\beta\delta)}{\alpha^2\varepsilon}\right)$ samples.*

## 5 Learning Mixtures of Axis-Aligned Gaussians

In this section, we prove the following result, which is a formal version of Theorem 1.1.

**Theorem 5.1.** *For any $\varepsilon \in (0, 1)$ and $\delta \in (0, 1/n)$, there is an $(\varepsilon, \delta)$-DP PAC learner for $k$-mix$\left(\mathcal{G}^d\right)$ and the number of samples used by the algorithm is $m(\alpha, \beta, \varepsilon, \delta) = \widetilde{O}\left(\frac{k^2 d \log^{3/2}(1/\beta\delta)}{\alpha^2\varepsilon}\right)$.*

The following lemma shows that we can construct an $(\varepsilon, \delta)$-DP list-decodable learner for the class of $d$-dimensional axis-aligned Gaussians, $\mathcal{G}^d$. We explain the high-level details of our approach here. Let $g = \prod_{i=1}^d g_i \in \mathcal{G}^d$ and $g' \in \mathcal{H}_\gamma(g)$. For a sample $X = (X_1, \ldots, X_d) \sim g'$, it follows that $X_i \sim g_i'$ where $g_i' \in \mathcal{H}_\gamma(g_i)$. We can thus split our dataset by dimension and run our univariate private list-decodable learner (Corollary 4.9) on each dimension separately to get a total of $d$ lists of univariate Gaussians. From the guarantee of Corollary 4.9, in the $i$th list there will be at least one Gaussian that is a good approximation to $g_i$, and this holds for all $i \in [d]$. Finally, since $g$ is an axis-aligned Gaussian (product distribution), we can take all possible combinations of the univariate distributions in the $d$ lists to obtain a new list of axis-aligned Gaussians, one of which accurately approximates $g$. The details of the proof can be found in Appendix E.

**Lemma 5.2.** *For any $\varepsilon \in (0,1)$ and $\delta \in (0, 1/n)$, there is an $(\varepsilon, \delta)$-DP L-list-decodable learner for $\mathcal{G}^d$ where $L = O\left(\frac{d^2}{(1-\gamma)^5 \alpha^2}\right)^d$ and the number of samples used by the algorithm is $m_{\mathrm{LIST}}(\alpha, \beta, \gamma, \varepsilon, \delta) = \widetilde{O}\left(\frac{d \log^{3/2}(1/\beta\delta)}{(1-\gamma)^2 \varepsilon}\right)$.*

We can now put together Lemma 5.2 and Theorem 3.1 to immediately get Theorem 5.1.

## Acknowledgments and Disclosure of Funding

CL was supported by an NSERC postdoctoral fellowship. HA was supported by an NSERC Discovery Grant.

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
