# A  Useful Facts

**Proposition A.1** (Lemma 2.11, [6])**.** *For any* $\mu, \widetilde{\mu} \in \mathbb{R}$ *and* $\sigma, \widetilde{\sigma} > 0$ *with* $|\widetilde{\mu} - \mu| \leq \alpha\sigma$ *and* $|\widetilde{\sigma} - \sigma| \leq \alpha\sigma$ *where* $\alpha \in [0, 2/3]$*, the Gaussians* $\mathcal{N}(\mu, \sigma^2)$ *and* $\mathcal{N}(\widetilde{\mu}, \widetilde{\sigma}^2)$ *statisfy*

$$d_{\mathrm{TV}}\left(\mathcal{N}(\mu, \sigma^2), \mathcal{N}(\widetilde{\mu}, \widetilde{\sigma}^2)\right) \leq \alpha.$$

**Proposition A.2** (Lemma 3.3.7, [57])**.** *For* $i \in [d]$ *let* $p_i$ *and* $q_i$ *be distributions over the same domain* $Z$*. Then*

$$d_{\mathrm{TV}}\left(\prod_{i=1}^{d} p_i, \prod_{i=1}^{d} q_i\right) \leq \sum_{i=1}^{d} d_{\mathrm{TV}}\left(p_i, q_i\right).$$

**Definition A.3** ($\alpha$-net)**.** *Let* $(X, d)$ *be a metric space. A set* $N \subseteq X$ *is an* $\alpha$-net *for* $X$ *under the metric* $d$ *if for all* $x \in X$*, there exists* $y \in N$ *such that* $d(x, y) \leq \alpha$*.*

**Proposition A.4.** *For any* $\alpha \in (0, 1]$ *and* $k \geq 2$*, there exists an* $\alpha$-net *of* $\Delta_k$ *under the* $\ell_\infty$-norm *of size at most* $(3/\alpha)^k$*.*

*Proof.* We will give an algorithmic proof of this fact. Let $r = \lceil 1/\alpha \rceil$ and fix $x \in \Delta_k$. Let $\ell = \sum_{i=1}^{k} rx_i - \lfloor rx_i \rfloor$. Note that $\sum_{i=1}^{k} rx_i = r$ and $rx_i - \lfloor rx_i \rfloor \in [0, 1)$ so $\ell$ is an integer in the interval $[0, r-1]$. Now define $\hat{x}$

$$\hat{x}_i = \begin{cases} \frac{\lfloor rx_i \rfloor + 1}{r} & i \leq \ell \\ \frac{\lfloor rx_i \rfloor}{r} & i > \ell \end{cases}.$$

Clearly, $\|x - \hat{x}\|_\infty \leq 1/r \leq \alpha$. It remains to check that $\hat{x} \in \Delta_k$. Indeed,

$$\sum_{i=1}^{k} \hat{x}_i = \sum_{i=1}^{k} \frac{\lfloor rx_i \rfloor}{r} + \frac{\ell}{r} = \sum_{i=1}^{k} \frac{\lfloor rx_i \rfloor}{r} + \sum_{i=1}^{k} \frac{rx_i - \lfloor rx_i \rfloor}{r} = 1,$$

where in the second equality, we used the definition of $\ell$. Note that for each $i$, $\hat{x}_i \in \{0, 1/r, 2/r, \ldots, 1\}$ so this shows that

$$\widehat{\Delta}_k = \{(t_1/r, \ldots, t_k/r) : t \in \mathbb{Z}_{\geq 0}^k, \|t\|_1 = r\},$$

is an $\alpha$-net for $\Delta_k$ of size $(r+1)^k$. To obtain the bound as asserted in the claim, note that $r + 1 = \lceil 1/\alpha \rceil + 1 \leq 1/\alpha + 2 \leq 3/\alpha$ for $\alpha \in (0, 1]$. $\qquad\square$

**Lemma A.5** (Chernoff bound; see [62, Exercise 2.3.6])**.** *Let* $X_1, \ldots, X_n$ *be independent Bernoulli random variables. Let* $S_n = \sum_{i=1}^{n} X_i$ *and* $\mu = S_n$*. Then for any* $\delta \in (0, 1]$ *and some absolute constant* $c > 0$

$$\mathbf{P}[|S_n - \mu| \geq \delta\mu] \leq 2e^{-c\mu\delta^2}.$$

# B  Locally Small Covers for Mixtures

To formally state and prove the impossibility result, we first introduce some useful definitions and results.

**Definition B.1** (TV ball)**.** *The total variation ball of radius* $\gamma \in (0, 1)$*, centered at a distribution* $g$ *with respect to a set of distributions* $\mathcal{F}$*, written* $\mathcal{B}(\gamma, g, \mathcal{F})$*, is the following subset of* $\mathcal{F}$*:*

$$\mathcal{B}(\gamma, g, \mathcal{F}) \coloneqq \{f \in \mathcal{F} : d_{\mathrm{TV}}(g, \mathcal{F}) \leq \gamma\}.$$

In this paper we consider coverings and packings of sets of distributions with respect to the total variation distance.

**Definition B.2** ($\gamma$-covers and $\gamma$-packings)**.** *For any* $\gamma \in (0, 1)$ *a* $\gamma$-cover *of a set of distributions* $\mathcal{F}$ *is a set of distributions* $\mathcal{C}_\gamma$*, such that for every* $f \in \mathcal{F}$*, there exists some* $\widehat{f} \in \mathcal{C}_\gamma$ *such that* $d_{\mathrm{TV}}(f, \widehat{f}) \leq \gamma$*.*

*A* $\gamma$-packing *of a set of distributions* $\mathcal{F}$ *is a set of distributions* $\mathcal{P}_\gamma \subseteq \mathcal{F}$*, such that for every pair of distributions* $f, f' \in \mathcal{P}_\gamma$*, we have that* $d_{\mathrm{TV}}(f, f') \geq \gamma$*.*

**Definition B.3** ($\gamma$-covering and $\gamma$-packing number). *For any $\gamma \in (0, 1)$, the $\gamma$-covering number of a set of distributions $\mathcal{F}$, $N(\mathcal{F}, \gamma) := \min\{n \in \mathbb{N} : \exists \mathcal{C}_\gamma \text{ s.t. } |\mathcal{C}_\gamma| = n\}$, is the size of the smallest possible $\gamma$-covering of $\mathcal{F}$. Similarly, the $\gamma$-packing number of a set of distributions $\mathcal{F}$, $M(\mathcal{F}, \gamma) := \max\{n \in \mathbb{N} : \exists \mathcal{P}_\gamma \text{ s.t. } |\mathcal{P}_\gamma| = n\}$, is the size of the largest subset of $\mathcal{F}$ that forms a packing for $\mathcal{F}$.*

The following Proposition follows directly from a well known relationship between packings and covers of metric spaces (see [62, Lemma 4.2.8]).

**Proposition B.4.** *For a set of distributions $\mathcal{F}$ with $\gamma$-covering number $M(\mathcal{F}, \gamma)$ and $\gamma$-packing number $N(\mathcal{F}, \gamma)$, the following holds:*

$$M(\mathcal{F}, 2\gamma) \leq N(\mathcal{F}, \gamma) \leq M(\mathcal{F}, \gamma).$$

We now formally define what it means for a set of distributions to be "locally small".

**Definition B.5** ($\gamma$-locally small). *Fix some $\gamma \in (0, 1)$. We say a set of distributions $\mathcal{F}$ is $\gamma$-locally small if*

$$\sup_{f \in \mathcal{F}} |\mathcal{B}(\gamma, f, \mathcal{F})| \leq k,$$

*for some $k \in \mathbb{N}$. If no such $k$ exists, we say $\mathcal{F}$ is not $\gamma$-locally small.*

**Proposition B.6.** *For every $\gamma \in (0, 1)$, any $(\gamma/2)$-cover for 2-mix$(\mathcal{G})$ is not $\gamma$-locally small.*

*Proof.* Fix some $\gamma \in (0, 1)$. Let $f = \mathcal{N}(0, 1)$ and define $g(\mu) := (1 - \gamma)\mathcal{N}(0, 1) + \gamma\mathcal{N}(\mu, 1)$ (note that $f = g(0)$). We will show that the following two statements hold for every $\mu, \mu' \in \mathbb{R}$:

1. $d_{\text{TV}}(g(\mu), g(\mu')) \leq \gamma$, and

2. If $|\mu - \mu'| \geq C$ for a sufficiently large constant $C$, $d_{\text{TV}}(g(\mu), g(\mu')) \geq \gamma/2$.

Consider the set of distributions $\mathcal{F} = \{g(\mu) : \mu \in \{C, 2C, \dots\}\}$ for some large positive constant $C$. For every $g, g' \in \mathcal{F}$, it follows from claim 1 that $g, g' \in \mathcal{B}(\gamma, f, 2\text{-mix}(\mathcal{G}))$ and from claim 2 that $d_{\text{TV}}(g, g') \geq \gamma/2$ for sufficiently large $C$. Thus, the $(\gamma/2)$-packing number of $\mathcal{B}(\gamma, f, 2\text{-mix}(\mathcal{G}))$ is unbounded, and by Proposition B.4, the $(\gamma/2)$-covering number of $\mathcal{B}(\gamma, f, 2\text{-mix}(\mathcal{G}))$ is also unbounded. This implies that *every* $(\gamma/2)$-cover for 2-mix$(\mathcal{G})$ is not $\gamma$-locally small by definition.

It remains to prove the two claims above. From the definition of the TV distance we have

$$
\begin{aligned}
d_{\text{TV}}(g(\mu), g(\mu')) &= \frac{1}{2} \|(1 - \gamma)\mathcal{N}(0, 1) + \gamma\mathcal{N}(\mu, 1) - (1 - \gamma)\mathcal{N}(0, 1) - \gamma\mathcal{N}(\mu', 1)\|_1 \\
&= \frac{\gamma}{2} \|\mathcal{N}(\mu, 1) - \mathcal{N}(\mu', 1)\|_1 \\
&= \gamma d_{\text{TV}}(\mathcal{N}(\mu, 1), \mathcal{N}(\mu', 1)).
\end{aligned}
\tag{1}
$$

Using the trivial upper bound on the TV distance between any two distributions, we have from Eq. (1) that $d_{\text{TV}}(g(\mu), g(\mu')) \leq \gamma$, which proves the first claim. If $|\mu - \mu'| \geq C$ for sufficiently large $C$, it follows from Gaussian tail bounds that $d_{\text{TV}}(\mathcal{N}(\mu, 1), \mathcal{N}(\mu', 1)) = 1 - \exp(-\Omega(C^2))$. Thus, by choosing $C$ to be sufficiently large, it follows from Eq. (1) that $d_{\text{TV}}(g(\mu), g(\mu')) \geq \gamma/2$. $\qquad\square$

## C   Omitted Proofs from Section 3

In this Appendix, we prove Theorem 3.1 which we restate here for convenience.

**Theorem 3.1.** *Let $k \in \mathbb{N}$ and $\varepsilon, \delta \in (0, 1)$. If $\mathcal{F}$ is $(\varepsilon/2, \delta)$-DP $L$-list-decodable with $m_{\text{LIST}}$ samples then there is an $(\varepsilon, \delta)$-DP PAC learner for $k$-mix$(\mathcal{F})$ where the number of samples used is*

$$
m(\alpha, \beta, \varepsilon, \delta) =
$$
$$
m_{\text{LIST}}\left(\frac{\alpha}{18}, \frac{\beta}{2k}, 1 - \frac{\alpha}{18k}, \frac{\varepsilon}{2}, \delta\right) + O\left(\frac{k\log(Lk/\alpha) + \log(1/\beta)}{\alpha^2} + \frac{k\log(Lk/\alpha) + \log(1/\beta)}{\alpha\varepsilon}\right).
$$

Algorithm 2 shows how a list-decodable learner can be used as a subroutine for learning mixture distributions. In the algorithm, we also make use of a subroutine for private hypothesis selection [4, 17]. In hypothesis selection, an algorithm is given i.i.d. sample access to some unknown distribution as well as a list of distributions to pick from. The goal of the algorithm is to output a distribution in the list that is close to the unknown distribution.

**Lemma C.1** ([4, Theorem 27]). *Let $n \in \mathbb{N}$. There exist an $(\varepsilon/2)$-DP algorithm $\mathrm{PHS}(\varepsilon, \alpha, \beta, \mathcal{F}, D)$ with the following property: for every $\varepsilon, \alpha, \beta \in (0, 1)$, and every set of distributions $\mathcal{F} = \{f_1, \ldots, f_M\}$, when PHS is given $\varepsilon, \alpha, \beta, \mathcal{F}$, and a dataset $D$ of $n$ i.i.d. samples from an unknown (arbitrary) distribution $g$ as input, it outputs a distribution $f_j \in \mathcal{F}$ such that*

$$d_{\mathrm{TV}}(g, f_j) \leq 3 \cdot d_{\mathrm{TV}}(g, \mathcal{F}) + \alpha/2,$$

*with probability no less than $1 - \beta/2$ so long as*

$$n = \Omega\left(\frac{\log(M/\beta)}{\alpha^2} + \frac{\log(M/\beta)}{\alpha\varepsilon}\right).$$

---

**Algorithm 2:** `Learn-Mixture`$(\alpha, \beta, \varepsilon, \delta, k, D)$.

**Input**   : Parameters $\alpha, \beta, \varepsilon, \delta > 0$, $k \in \mathbb{N}$ and dataset $D$ of $n$ i.i.d. samples generated $g$.

**Output** : mixture $\widehat{g} = \sum_{i=1}^{n} \widehat{w}_i \widehat{f}_i$.

1 Split $D$ into $D_1, D_2$ where $|D_1| = n_1$, $|D_2| = n - n_1$ // $n_1 = m_{\mathtt{List}}\left(\frac{\varepsilon}{2}, \delta, \frac{\alpha}{18}, \frac{\beta}{2k}, 1 - \frac{\alpha}{18k}\right)$.

2 $\widehat{\mathcal{F}} = \{\widehat{f}_1, \ldots, \widehat{f}_L\} \leftarrow \mathcal{A}_{\mathrm{LIST}}(\alpha/18, \beta/2k, 1 - \alpha/18k, \varepsilon/2, \delta, D_1)$ // $\left(\frac{\varepsilon}{2}, \delta\right)$-DP
   $L$-`list-decodable learner`.

3 Set $\widehat{\Delta}_k$ as $(18k/\alpha)$-net of $\Delta_k$ from Proposition A.4

4 Set $\mathcal{K} = \{\sum_{i=1}^{k} \widehat{w}_i \widehat{f}_i : \widehat{w} \in \widehat{\Delta}_k, \widehat{f}_i \in \widehat{\mathcal{F}}\}$

5 $\widehat{g} \leftarrow \mathrm{PHS}(\varepsilon/2, \alpha, \beta/2, \mathcal{K}, D_2)$

6 **Return** $\widehat{g}$

---

*Proof of Theorem 3.1.* We begin by briefly showing that Algorithm 2 satisfies $(\varepsilon, \delta)$-DP before arguing about its utility.

**Privacy.**   We first prove that Algorithm 2 is $(\varepsilon, \delta)$-DP. Step 2 of the algorithm satisfies $(\varepsilon/2, \delta)$-DP by the fact that $\mathcal{A}_{\mathrm{List}}$ is an $(\varepsilon/2, \delta)$-DP $L$-list-decodable learner. Steps 3 and 4 maintain $(\varepsilon/2, \delta)$-DP by post processing (Lemma 2.9). Finally, step 5 satisfies $(\varepsilon/2)$-DP by Lemma C.1. By basic composition (Lemma 2.8) the entire algorithm is $(\varepsilon, \delta)$-DP.

**Utility.**   We now proceed to show that Algorithm 2 PAC learns $k$-mix$(\mathcal{F})$. In step 2 of Algorithm 2, we use the $(\varepsilon/2, \delta)$-DP $L$-list-decodable learner to obtain a set of distributions $\widehat{\mathcal{F}}$ of size at most $L$. Note that for any mixture component $f_j$, $g$ is a $(1 - w_j)$-corrupted distribution of $f_j$ since

$$g = w_j f_j + \sum_{i \neq j} w_i f_i = w_j f_j + (1 - w_j) \sum_{i \neq j} \frac{w_i f_i}{1 - w_j} = w_j f_j + (1 - w_j) h,$$

where $h = \sum_{i \neq j} \frac{w_i f_i}{1 - w_j}$.

Let $N = \{i \in [k] : w_i \geq \alpha/18k\}$ denote the set of *non-negligible* components. We first show that for any non-negligible component $i \in N$, there exists $\widehat{f} \in \widehat{\mathcal{F}}$ that is close to $f_i$.

**Claim C.2.** *If $|D_1| \geq m_{\mathrm{LIST}}(\alpha/18, \beta/2k, 1 - \alpha/18k, \varepsilon/2, \delta)$ then $d_{\mathrm{TV}}(f_i, \widehat{\mathcal{F}}) \leq \alpha/18$ for all $i \in N$ with probability at least $1 - \beta/2$.*

*Proof.* Fix $i \in N$. Note that $1 - w_i \leq 1 - \alpha/18k$ so $f \in \mathcal{H}_{1 - \alpha/18k}(f_i)$. Since step 2 of Algorithm 2 makes use of a list-decodable learner, as long as $|D_1| \geq m_{\mathrm{LIST}}(\alpha/18, \beta/2k, 1 - \alpha/18k, \varepsilon/2, \delta)$ we have $d_{\mathrm{TV}}(f_i, \widehat{\mathcal{F}}) \leq \alpha/18$ with probability at least $1 - \beta/2k$. Since this is true for any fixed $i \in N$, a union bound gives that $d_{\mathrm{TV}}(f_i, \widehat{\mathcal{F}}) \leq \alpha/18$ for all $i \in N$ with probability at least $1 - \beta/2$. $\square$

Steps 3 and 4 of Algorithm 2 constructs a candidate set $\mathcal{K}$ of mixture distributions using $\widehat{\mathcal{F}}$ and a net of the probability simplex $\Delta_k$. The next claim shows that as long as $d_{\mathrm{TV}}(f_i, \widehat{\mathcal{F}})$ is small for every non-negligible $i \in N$, $d_{\mathrm{TV}}(g, \mathcal{K})$ is small as well.

**Claim C.3.** *If $d_{\mathrm{TV}}(f_i, \widehat{\mathcal{F}}) \leq \alpha/18$ for every $i \in N$, then $d_{\mathrm{TV}}(g, \mathcal{K}) \leq \alpha/6$. In addition, $|\mathcal{K}| \leq \left(\frac{54Lk}{\alpha}\right)^k$.*

*Proof.* Step 3 constructs a set $\widehat{\Delta}_k$ which is an $(18k/\alpha)$-net of the probability simplex $\Delta_k$ in the $\ell_\infty$-norm. By the hypothesis of the claim, for each $i \in N$, there exists $\widehat{f}_i \in \widehat{\mathcal{F}}$ such that $d_{\mathrm{TV}}(f_i, \widehat{f}_i) \leq \alpha/18$. Recall that $g = \sum_{i \in [k]} w_i f_i$. Let $\widehat{w} \in \widehat{\Delta}_k$ such that $\|\widehat{w} - w\|_\infty \leq \alpha/18k$. Now let $\widetilde{g} = \sum_{i \in [k]} \widehat{w}_i \widehat{f}_i$. Note that $\widetilde{g} \in \mathcal{K}$. Moreover, a straightforward calculation shows that $d_{\mathrm{TV}}(g, \widetilde{g}) \leq \alpha/6$ (see Proposition C.4 for the detailed calculations). This proves that $d_{\mathrm{TV}}(g, \mathcal{K}) \leq \alpha/6$.

Lastly, to bound $|\mathcal{K}|$ we have $|\mathcal{K}| \leq |\widehat{\mathcal{F}}|^k \cdot |\widehat{\Delta}_k|$. Note that $|\widehat{\mathcal{F}}| \leq L$ since it is the output of an $L$-list-decodable learner and $|\widehat{\Delta}_k| \leq (54k/\alpha)^k$ by Proposition A.4. This implies the claimed bound on $|\mathcal{K}|$. $\qquad\square$

The only remaining step is to select a good hypothesis from $\mathcal{K}$. This is achieved using the private hypothesis selection algorithm from Lemma C.1 which guarantees that step 5 of Algorithm 2 returns $\widehat{g}$ satisfying $d_{\mathrm{TV}}(g, \widehat{g}) \leq 3 \cdot d_{\mathrm{TV}}(g, \mathcal{K}) + \alpha/2$ with probability $1 - \beta/2$ as long as

$$|D_2| = \Omega\left(\frac{\log(|\mathcal{K}|/\beta)}{\alpha^2} + \frac{\log(|\mathcal{K}|/\beta)}{\alpha\varepsilon}\right) = \Omega\left(\frac{k\log(Lk/\alpha) + \log(1/\beta)}{\alpha^2} + \frac{k\log(Lk/\alpha) + \log(1/\beta)}{\alpha\varepsilon}\right).$$
(2)

Combining this with Claim C.2, Claim C.3, and a union bound, we have that with probability $1 - \beta$,

$$d_{\mathrm{TV}}(g, \widehat{g}) \leq 3 \cdot d_{\mathrm{TV}}(g, \mathcal{K}) + \alpha/2 \leq \alpha,$$

where the first inequality follows from private hypothesis selection and the second inequality follows from Claim C.2 and Claim C.3.

Finally, the claimed sample complexity bound follows from the samples required to construct $\widehat{\mathcal{F}}$ (which follows from Claim C.2) and the samples required for private hypothesis selection which is given in Eq. (2). $\qquad\square$

**Proposition C.4.** *Let $\alpha \in (0, 1)$ and $k \in \mathbb{N}$. Let $g = \sum_{i=1}^k w_i f_i$ and $\widehat{g} = \sum_{i=1}^k \widehat{w}_i \widehat{f}_i$ be two mixture distributions that satisfy*

1. *$\|w - \widehat{w}\|_\infty \leq \alpha/k$; and*

2. *$d_{\mathrm{TV}}(f_i, \widehat{f}_i) \leq \alpha$ for $i \in [k]$ such that $w_i \geq \alpha/k$.*

*Then $d_{\mathrm{TV}}(\widehat{g}, g) \leq 3\alpha$.*

*Proof.* Let $N = \{i \in [k] : w_i \geq \alpha/k\}$. We have that

$$
\begin{aligned}
d_{\mathrm{TV}}(\widehat{g}, g) &= \frac{1}{2}\left\|\sum_{i=1}^k \widehat{w}_i \widehat{f}_i - \sum_{i=1}^k w_i f_i\right\|_1 \\
&= \frac{1}{2}\left\|\sum_{i=1}^k \widehat{w}_i(\widehat{f}_i - f_i) + \sum_{i=1}^k (\widehat{w}_i - w_i) f_i\right\|_1 \\
&\leq \frac{1}{2}\left\|\sum_{i=1}^k \widehat{w}_i(\widehat{f}_i - f_i)\right\|_1 + \frac{1}{2}\left\|\sum_{i=1}^k (\widehat{w}_i - w_i) f_i\right\|_1 \\
&\leq \frac{1}{2}\left\|\sum_{i\notin N} \widehat{w}_i(\widehat{f}_i - f_i)\right\|_1 + \frac{1}{2}\left\|\sum_{i\in N} \widehat{w}_i(\widehat{f}_i - f_i)\right\|_1 + \frac{1}{2}\left\|\sum_{i=1}^k (\widehat{w}_i - w_i) f_i\right\|_1
\end{aligned}
$$

$$\leq \frac{1}{2} \sum_{i \notin N} \widehat{w}_i \left\| \widehat{f}_i - f_i \right\|_1 + \frac{1}{2} \sum_{i \in N} \widehat{w}_i \left\| \widehat{f}_i - f_i \right\|_1 + \frac{1}{2} \sum_{i=1}^{k} |\widehat{w}_i - w_i| \left\| \widehat{f}_i \right\|_1$$

$$\leq \sum_{i \notin N} \frac{\alpha}{k} \cdot 1 + \sum_{i \in N} \widehat{w}_i \cdot \alpha + \sum_{i=1}^{k} \frac{\alpha}{k} \cdot 1$$

$$\leq \alpha + \alpha + \alpha = 3\alpha.$$

Note that in the second-to-last inequality, we used that for $i \notin N$, $\widehat{w}_i \leq \alpha/k$ and the trivial bound $\|\widehat{f}_i - f_i\|_1 \leq 2$ while for $i \in N$, we have $\|\widehat{f}_i - f_i\|_1 \leq \alpha$. □

## D   Omitted Results from Section 4

### D.1   Proofs of Claim 4.3, Claim 4.4, and Corollary 4.5

*Proof of Claim 4.3.* First, observe that for a bin $B_i = ((i - 0.5)\widetilde{\sigma}, (i + 0.5)\widetilde{\sigma}]$ and $X \sim g'$, we have (recalling Definition 2.4), $p_i = \mathbf{P}_{X \sim g'}[X \in B_i] \geq (1 - \gamma)\mathbf{P}_{X \sim g}[X \in B_i]$. A fairly straightforward calculation (see Proposition D.5) gives that $\mathbf{P}_{X \sim g}[X \in B_j] \geq 1/3$ so that $p_j \geq (1 - \gamma)/3$.

A standard Chernoff bound (Lemma A.5) implies that $|\overline{p}_j - p_j| < p_j/2$ with probability at least $1 - \beta/2$ provided $n \geq C \log(1/\beta)/(1 - \gamma)$ for some constant $C > 0$. As $p_j \geq (1 - \gamma)/3$ this implies $\overline{p}_j > (1 - \gamma)/6$. □

*Proof of Claim 4.4.* The first assertion directly follows from Lemma 4.1 with $\eta = (1 - \gamma)/24$. In the event that $|\overline{p}_i - \widetilde{p}_i| \leq (1 - \gamma)/24$, we now show that $|H| \leq 12/(1 - \gamma)$. Note that it suffices to argue that if $i \in H$ then $\overline{p}_i > (1 - \gamma)/12$. Since $\sum_{i \in \mathbb{N}} \overline{p}_i = 1$, this implies that $|H| \leq 12/(1 - \gamma)$. Indeed, we argue the contrapositive. If $\overline{p}_i \leq (1 - \gamma)/12$ then $\widetilde{p}_i \leq \overline{p}_i + (1 - \gamma)/24 \leq (1 - \gamma)/8$ and, hence, $i \notin H$. □

*Proof of Corollary 4.5.* The algorithm is simple; we run $\mathtt{Univariate\text{-}Mean\text{-}Decoder}(\varepsilon, \delta, \beta, \gamma, \sigma, D)$ and obtain the set $\widetilde{M}$. Let $\widehat{M}$ be an $\alpha\sigma$-net of the set of intervals $\{[\widetilde{\mu} - \sigma, \widetilde{\mu} + \sigma] : \widetilde{\mu} \in \widetilde{M}\}$ of size $|\widetilde{M}| \cdot (2 \cdot \lceil 1/2\alpha \rceil + 1)$, i.e.

$$\widehat{M} = \{\widetilde{\mu} + 2j\alpha\sigma : \widetilde{\mu} \in \widetilde{M}, \, j \in \{0, \pm 1, \ldots, \pm\lceil 1/2\alpha \rceil\}\}.$$

We then return $\widehat{\mathcal{F}} = \{\mathcal{N}(\widehat{\mu}, \sigma^2) : \widehat{\mu} \in \widehat{M}\}$. Finally, Lemma 4.2 and post-processing (Lemma 2.9) imply that the algorithm is $(\varepsilon, \delta)$-DP while Lemma 4.2 and Proposition A.1 imply the accuracy guarantee.[4] □

### D.2   Proof of Lemma 4.7

The algorithm for estimating the variance is given in Algorithm 3. The rest of this subsection makes reference to that algorithm.

Let $g = \mathcal{N}(\mu, \sigma^2)$ and $g' \in \mathcal{H}_\gamma(g)$. Let $X, X' \sim g'$ and let $Y = |X - X'|/\sqrt{2}$. For an integer $i$, let $p_i = \mathbf{P}[Y \in B_i]$ where $B_i = (2^i, 2^{i+1}]$. Let $j$ be the (unique) integer such that $\sigma \in (2^j, 2^{j+1}]$.

**Claim D.1.** *If $n = \Omega(\log(1/\beta)/(1 - \gamma)^2)$ then $\overline{p}_j > (1 - \gamma)^2/6$ with probability $1 - \beta/2$.*

*Proof.* Since, $X, X' \sim g'$ and $Y = |X - X'|/\sqrt{2}$, a straightforward calculation shows that $p_j \geq (1 - \gamma)^2/4$ (see Proposition D.6 and Proposition D.7 for details).

Next, a standard Chernoff bound (Lemma A.5) implies that $|\overline{p}_j - p_j| < p_j/3$ with probability at least $1 - \beta/2$ provided $n \geq C \log(1/\beta)/(1 - \gamma)^2$ for some constant $C > 0$. As $p_j \geq (1 - \gamma)^2/4$ this implies $\overline{p}_j > (1 - \gamma)^2/6$. □

---

[4]Note that we can only use Proposition A.1 for target $\alpha$ as large as $2/3$. For any target $\alpha > 2/3$, we can simply run the algorithm with $\alpha = 2/3$.

---

**Algorithm 3:** `Univariate-Variance-Decoder`$(\beta, \gamma, \varepsilon, \delta, D)$.

---

**Input** : Parameters $\varepsilon, \beta, \gamma \in (0,1)$, $\delta \in (0, 1/n)$, and a dataset $D$

**Output:** Set of approximate standard deviations $\widetilde{V} = \{\widetilde{\sigma}_1, \ldots, \widetilde{\sigma}_L\}$.

1 $Y_k \leftarrow |(X^{2k} - X^{2k-1})/\sqrt{2}|$ for $k \in [n]$.      // $X^i$s from Dataset $D = \{X^1, \ldots, X^{2n}\}$

2 $D' \leftarrow \{Y_1, \ldots, Y_n\}$.

3 Partition $\mathbb{R}_{>0}$ into bins $\mathbf{B} = \{B_i\}_{i \in \mathbb{Z}}$ where $B_i = (2^i, 2^{i+1}]$.

4 $\{\widetilde{p}_i\}_{i \in \mathbb{Z}} \leftarrow$ `Stable-Histogram`$(\varepsilon, \delta, (1-\gamma)^2/24, \beta/2, D', \mathbf{B})$.

5 $H \leftarrow \{i : \widetilde{p}_i > (1-\gamma)^2/8\}$

6 If $|H| > 12/(1-\gamma)^2$ **fail** and return $\widetilde{V} = \emptyset$

7 $\widetilde{V} \leftarrow \{2^{i+1} : i \in H\}$.

8 **Return** $\widetilde{V}$

---

**Claim D.2.** *If $n = \Omega(\log(1/\beta\delta)/(1-\gamma)^2\varepsilon)$ then with probability $1 - \beta/2$, we have (i) $|\overline{p}_i - \widetilde{p}_i| \leq (1-\gamma)^2/24$ for all $i \in \mathbb{N}$ and (ii) $|H| = |\{i \in \mathbb{N} : \widetilde{p}_i > (1-\gamma)^2/8\}| \leq 12/(1-\gamma)^2$.*

*Proof.* The first assertion directly follows from Lemma 4.1 with $\eta = (1-\gamma)^2/24$. In the event that $|\overline{p}_i - \widetilde{p}_i| \leq (1-\gamma)^2/24$, we now show that $|H| \leq 12/(1-\gamma)^2$. Note that it suffices to argue that if $i \in H$ then $\overline{p}_i > (1-\gamma)^2/12$. Since $\sum_{i \in \mathbb{N}} \overline{p}_i = 1$, this implies that $|H| \leq 12/(1-\gamma)^2$. Indeed, we argue the contrapositive. If $\overline{p}_i \leq (1-\gamma)^2/12$ then $\widetilde{p}_i \leq \overline{p}_i + (1-\gamma)^2/24 \leq (1-\gamma)^2/12$ and, hence, $i \notin H$. $\qquad\square$

Given Claim D.1 and Claim D.2, we now prove Lemma 4.7.

*Proof of Lemma 4.7.* We briefly prove that the algorithm is private before proceeding to the other assertions of the lemma.

**Privacy.** Line 4 is the only part of the algorithm that looks at the data and it is $(\varepsilon, \delta)$-DP by Lemma 4.1. The remainder of the algorithm can be viewed as post-processing (Lemma 2.9) so does not affect the privacy.

**Bound on $|\widetilde{V}|$.** For the bound on $|\widetilde{V}|$, observe that if $|H| > 12/(1-\gamma)^2$ then the algorithm fails so $|\widetilde{V}| \leq 12/(1-\gamma)^2$ deterministically.

**Utility.** Let $g, g', \sigma$ be as defined in the statement of the lemma. We now show that there exists $\widetilde{\sigma} \in \widetilde{V}$ such that $\widetilde{\sigma} \in [\sigma, 2\sigma)$. Let $j$ be the unique integer such that $\sigma \in (2^j, 2^{j+1}]$. For the remainder of the proof, we assume that $n = \Omega(\log(1/\beta\delta)/(1-\gamma)^2\varepsilon)$.

Claim D.1 asserts that, with probability $1 - \beta/2$, we have $\overline{p}_j > (1-\gamma)^2/6$. Claim D.2 asserts that, with probability $1 - \beta/2$, $\widetilde{p}_j \geq \overline{p}_j - (1-\gamma)^2/24$ *and* that $|H| \leq 12/(1-\gamma)^2$. By a union bound, with probability $1 - \beta$, we have that $\overline{p}_j > (1-\gamma)^2/8$ and the algorithm does not fail. This implies that $j \in H$ so $2^{j+1} \in \widetilde{V}$ and, by the choice of $j$, $\sigma \leq 2^{j+1} < 2\sigma$. This completes the proof. $\qquad\square$

### D.3 Proof of Lemma 4.8

The algorithm for estimating the variance is given in Algorithm 4. The rest of this subsection makes reference to that algorithm.

Before we prove the lemma, we make a few simple observations. Fix $g = \mathcal{N}(\mu, \sigma^2)$ and $g' \in \mathcal{H}_\gamma(g)$. We assume that the algorithm receives $D \sim (g')^{2n}$ as input.

**Claim D.3.** *If $n_1 = \Omega(\log(1/\beta\delta)/(1-\gamma)^2\varepsilon)$ then with probability $1 - \beta/2$, (i) there exists $\widetilde{\sigma} \in \widetilde{V}$ such that $\widetilde{\sigma} \in [\sigma, 2\sigma)$ and (ii) there exists $\widehat{\sigma} \in \widehat{V}$ such that $|\widehat{\sigma} - \sigma| \leq \alpha\sigma$.*

*Proof.* Lemma 4.7 directly implies that in line 4, with probability $1 - \beta/2$, there is some $\widetilde{\sigma} \in \widetilde{V}$ such that $\widetilde{\sigma} \in [\sigma, 2\sigma)$.

---
**Algorithm 4:** `Univariate-Gaussian-Decoder`$(\alpha, \beta, \gamma, \varepsilon, \delta, D)$.
---
**Input** : Parameters $\varepsilon, \alpha, \beta, \gamma \in (0, 1)$, $\delta \in (0, 1/n)$ and a dataset $D$

**Output** : Set of approximate means $\widehat{M}$ and variances $\widehat{V}$.

1 Set $T = 12/(1 - \gamma)^2$

2 Set $\varepsilon' = \varepsilon/(2\sqrt{6T \log(2(T+1)/\delta)})$ and $\delta' = \delta/2(T+1)$

3 Split $D$ into $D_1, D_2$ where $|D_1| = n_1$, $|D_2| = n_2 = n - n_1$
   $\quad$ // $n_1 = \Theta(\log(1/\beta\delta)/(1-\gamma)^2\varepsilon)$ .

4 $\widetilde{V} \leftarrow$ `Univariate-Variance-Decoder`$(\beta/2, \gamma, \varepsilon/2, \delta/2, D_1)$

5 Initialize $\widehat{M} \leftarrow \emptyset$

6 For $\widetilde{\sigma}_i \in \widetilde{V}$ **do**

7 $\quad \widetilde{M}_i =$ `Univariate-Mean-Decoder`$(\beta/2, \gamma, \varepsilon', \delta', \widetilde{\sigma}_i, D_2)$

8 $\quad \widehat{M}_i \leftarrow \{\widetilde{\mu} + j\alpha\widetilde{\sigma}_i : \widetilde{\mu} \in \widetilde{M}_i, j \in \{0, \pm1, \pm2, \ldots, \pm\lceil 1/\alpha\rceil\}$

9 $\quad \widehat{M} \leftarrow \widehat{M} \cup \widehat{M}_i$

10 $C \leftarrow \{\log_2(1+\alpha), 2\log_2(1+\alpha), \ldots, \lceil 1/\log_2(1+\alpha)\rceil \cdot \log_2(1+\alpha)\}$

11 $\widehat{V} \leftarrow \{\widetilde{\sigma} \cdot 2^{c-1} : \widetilde{\sigma} \in \widetilde{V}, c \in C\}$

12 **Return** $\widehat{M}, \widehat{V}$
---

For the final assertion, suppose that $\widetilde{\sigma} \in [\sigma, 2\sigma)$. In particular, $\log_2(2\sigma/\widetilde{\sigma}) \in (0, 1]$. Note that $C$ is $\log_2(1+\alpha)$-net of the interval $[0, 1]$. Hence, there exists some $c \in C$ such that $|c - \log_2(2\sigma/\widetilde{\sigma})| \leq \log_2(1+\alpha)$. For such a value of $c$, we have $(\widetilde{\sigma}/\sigma) \cdot 2^{c-1} \in [1/(1+\alpha), 1+\alpha]$, which upon rearranging gives $\widetilde{\sigma}2^{c-1} \in [\sigma/(1+\alpha), \sigma(1+\alpha)]$. As $1/(1+\alpha) \geq 1-\alpha$, this shows that $|\widetilde{\sigma}2^{c-1} - \sigma| \leq \alpha\sigma$. This completes the proof since $\widetilde{\sigma}2^{c-1} \in \widehat{V}$. $\qquad\square$

**Claim D.4.** *Let $\varepsilon', \delta'$ be as defined in Algorithm 4. Suppose that there exists $\widetilde{\sigma}_i \in \widetilde{V}$ such that $\widetilde{\sigma}_i \in [\sigma, 2\sigma)$. If $n_2 = \Omega(\log(1/\beta\delta')/(1-\gamma)\varepsilon')$ then with probability $1 - \beta/2$ there exists $\widehat{\mu} \in \widehat{M}$ such that $|\widehat{\mu} - \mu| \leq \alpha\sigma$.*

*Proof.* The condition that there exists $\widetilde{\sigma}_i \in \widetilde{V}$ such that $\widetilde{\sigma}_i \in [\sigma, 2\sigma)$ implies that one of the runs of `Univariate-Mean-Decoder` on line 7 uses $\widetilde{\sigma}_i \in [\sigma, 2\sigma)$. The guarantee of Lemma 4.2 shows that with probability $1 - \beta/2$, there is some $\widetilde{\mu} \in \widetilde{M}_i$ satisfying $|\widetilde{\mu} - \mu| \leq \sigma$. Finally, on line 8, the algorithm constructs $\widehat{M}_i$ which is a $(\alpha\widetilde{\sigma}_i/2)$-net of the interval $[\widetilde{\mu} - \widetilde{\sigma}_i, \widetilde{\mu} + \widetilde{\sigma}_i] \supset [\widetilde{\mu} - \sigma, \widetilde{\mu} + \sigma]$. Hence, there exists $\widehat{\mu} \in \widehat{M}_i$ such that $|\widehat{\mu} - \mu| \leq \alpha\widetilde{\sigma}/2 < \alpha\sigma$ where the latter inequality used that $\widetilde{\sigma} < 2\sigma$. Since $\widehat{M}_i \subset \widehat{M}$, this implies the claim. $\qquad\square$

*Proof of Lemma 4.8.* The list-decoding algorithm for univariate Gaussians is given in Algorithm 4.

**Privacy.** We first prove that the algorithm is $(\varepsilon, \delta)$-DP. By Lemma 4.2, line 4 satisfies $(\varepsilon/2, \delta/2)$-DP. The loop on line 6 runs at most $12/(1-\gamma)^2$ times since $|\widetilde{V}| \leq 12/(1-\gamma)^2$ (see Lemma 4.7). So, by our choice of $\varepsilon'$, $\delta'$ (line 2) and advanced composition (Lemma 2.8), all the iterations of line 7 collectively satisfy $(\varepsilon/2, \delta/2)$-DP. No subsequent part of the algorithm accesses the data so by basic composition (Lemma 2.8) and post processing (Lemma 2.9), the entire algorithm is $(\varepsilon, \delta)$-DP.

**Bound on $|\widehat{M}|$ and $|\widehat{V}|$.** We now prove the claimed upper bounds on the sizes of $\widehat{M}$ and $\widehat{V}$. First, we have $|\widetilde{V}| \leq 12/(1-\gamma)^2$ by Lemma 4.7. Since $|C| = \lceil 1/\log_2(1+\alpha)\rceil = \lceil \log_{1+\alpha}(2)\rceil$, this gives $|\widehat{V}| = |\widetilde{V}| \cdot |C| \leq 12 \cdot \lceil \log_{1+\alpha}(2)\rceil/(1-\gamma)^2$. Next, we have that each $|\widetilde{M}_i| \leq 12/(1-\gamma)$ in Line 8 by Lemma 4.2, so $|\widehat{M}_i| \leq 12 \cdot (2 \cdot \lceil 1/\alpha\rceil + 1)/(1-\gamma)$. Hence, $|\widehat{M}| \leq |\widetilde{V}| \cdot 12 \cdot (2 \cdot \lceil 1/\alpha\rceil + 1)/(1-\gamma) \leq 144 \cdot (2 \cdot \lceil 1/\alpha\rceil + 1)/(1-\gamma)^3$.

**Existence of $\widehat{\mu}$ and $\widehat{\sigma}$.** Claim D.3 asserts that with probability $1 - \beta/2$, there is $\widetilde{\sigma} \in \widetilde{V}$ such that $\widetilde{\sigma} \in [\sigma, 2\sigma)$ and that there exists $\widehat{\sigma} \in \widehat{V}$ such that $|\widehat{\sigma} - \sigma| \leq \alpha\sigma$. The latter statement is the bound that we asserted for $\widehat{\sigma}$ in the statement of the lemma.

Next, conditioning on the event that there exists $\widetilde{\sigma} \in \widetilde{V}$ such that $\widetilde{\sigma} \in [\sigma, 2\sigma)$, Claim D.4 implies that with probability $1 - \beta/2$, there is some $\widehat{\mu} \in \widehat{M}$ such that $|\widehat{\mu} - \mu| \leq \alpha\sigma$.

To conclude, taking a union bound shows that with probability $1 - \beta$, there exists $\widehat{\mu} \in \widehat{M}, \widehat{\sigma} \in \widehat{V}$ satisfying $|\widehat{\mu} - \mu| \leq \alpha\sigma$ and $|\widehat{\sigma} - \sigma| \leq \alpha\sigma$.

**Sample complexity.** Finally, we argue about the sample complexity. For Claim D.3, we needed $n_1 = \Omega(\log(1/\beta\delta)/(1-\gamma)^2\varepsilon)$ samples and for Claim D.4, we needed $n_2 = \Omega(\log(1/\beta\delta')/(1-\gamma)\varepsilon')$ samples. Adding $n_1, n_2$ and plugging in the values for $\varepsilon', \delta'$ as defined in Algorithm 4 gives the claimed bound on the number of samples required. $\qquad\square$

### D.4 Proof of Corollary 4.9

*Proof of Corollary 4.9.* We run `Univariate-Gaussian-Decoder`$(\alpha, \beta, \varepsilon, \delta, \gamma, D)$ and obtain the sets $\widehat{M}$ and $\widehat{V}$. We then output $\widehat{\mathcal{F}} = \{\mathcal{N}(\widehat{\mu}, \widehat{\sigma}) \; : \; \widehat{\mu} \in \widehat{M}, \; \widehat{\sigma} \in \widehat{V}\}$. The algorithm is $(\varepsilon, \delta)$-DP by the guarantee of Lemma 4.8 and post processing (Lemma 2.9). We have from the guarantee of Lemma 4.8 that

$$|\widehat{\mathcal{F}}| = |\widehat{M}| \cdot |\widehat{V}| \leq \left(\frac{1728}{(1-\gamma)^5}\right) \cdot \lceil \log_{1+\alpha}(2) \rceil \cdot (2\lceil 1/\alpha \rceil + 1).$$

Note that $\log_{1+\alpha}(2) = \frac{\ln(2)}{\ln(1+\alpha)} \leq \frac{2\ln(2)}{\alpha}$ where the last inequality follows from the inequality $\ln(1 + x) \geq x/2$ valid for $x \in [0, 1]$. This gives the claimed bound that $L = |\widehat{\mathcal{F}}| = O\left(\frac{1}{(1-\gamma)^5\alpha^2}\right)$.

For any $g \in \mathcal{G}$ and $g' \in \mathcal{H}_\gamma(g)$, given $n$ samples from $g'$ as input, we have from the guarantee of Lemma 4.8 and Proposition A.1 that the algorithm outputs $\widehat{\mathcal{F}}$ satisfying $d_{\mathrm{TV}}(g, \widehat{\mathcal{F}}) \leq \alpha$ so long as

$$n = \Omega\left(\frac{\log(1/\beta\delta)}{(1-\gamma)^2\varepsilon} + \frac{\log(1/(1-\gamma)\beta\delta)\sqrt{\log(1/(1-\gamma)\delta)}}{(1-\gamma)^2\varepsilon}\right) = \widetilde{\Omega}\left(\frac{\log^{3/2}(1/\beta\delta)}{(1-\gamma)^2\varepsilon}\right).$$

This proves the corollary. $\qquad\square$

### D.5 Useful facts

**Proposition D.5.** *Fix some univariate Gaussian $g = \mathcal{N}(\mu, \sigma^2)$. Let $\widetilde{\sigma}$ satisfy $\sigma \leq \widetilde{\sigma} < 2\sigma$. Partition $\mathbb{R}$ into disjoint bins $\{B_i\}_{i\in\mathbb{N}}$ where $B_i = ((i - 0.5)\widetilde{\sigma}, (i + 0.5)\widetilde{\sigma}]$ and let $j = \lceil \mu/\widetilde{\sigma} \rfloor$, where $\lceil \cdot \rfloor$ denotes rounding to the nearest integer. It follows that:*

1. $\mathbf{P}_{X\sim g}[X \in B_j] \geq 1/3$,
2. $\mu \in [(j - 0.5)\widetilde{\sigma}, (j + 0.5)\widetilde{\sigma}]$.

*Proof.* We first prove item 1.

$$\begin{aligned}
\mathbf{P}_{X\sim g}[X \in B_j] &= \Phi\left(\frac{(j + 0.5)\widetilde{\sigma}}{\sigma} - \frac{\mu}{\sigma}\right) - \Phi\left(\frac{(j - 0.5)\widetilde{\sigma}}{\sigma} - \frac{\mu}{\sigma}\right) \\
&= \Phi\left(\frac{j\widetilde{\sigma} - \mu}{\sigma} + \frac{\widetilde{\sigma}}{2\sigma}\right) - \Phi\left(\frac{j\widetilde{\sigma} - \mu}{\sigma} - \frac{\widetilde{\sigma}}{2\sigma}\right) \\
&:= f\left(\frac{j\widetilde{\sigma} - \mu}{\sigma}\right).
\end{aligned}$$

Notice that $f(\xi) = \Phi(\xi + \widetilde{\sigma}/2\sigma) - \Phi(\xi - \widetilde{\sigma}/2\sigma)$ is decreasing with $|\xi|$. Furthermore, by the definition of $j$ we have,

$$\left|\frac{j\widetilde{\sigma} - \mu}{\sigma}\right| = \frac{\widetilde{\sigma}}{\sigma}\left|j' - \frac{\mu}{\widetilde{\sigma}}\right|$$

$$\leq \frac{\widetilde{\sigma}}{\sigma} \cdot \frac{1}{2} = \frac{\widetilde{\sigma}}{2\sigma}.$$

So,

$$\mathbf{P}_{X\sim g}[X \in B_j] = f\left(\frac{j\widetilde{\sigma} - \mu}{\sigma}\right)$$

$$\geq f\left(\frac{\widetilde{\sigma}}{2\sigma}\right)$$

$$= \Phi\left(\frac{\widetilde{\sigma}}{\sigma}\right) - \Phi(0)$$

$$\geq \Phi(1) - \Phi(0) \geq 1/3,$$

where the second last inequality follows from the fact that $\widetilde{\sigma}/\sigma \geq 1$ together with the monotonicity of the c.d.f. and the last inequality follows from a direct calculation.

We now prove the second claim that $\mu \in [(j - 0.5)\widetilde{\sigma}, (j + 0.5)\widetilde{\sigma})]$. As we saw above, it follows that

$$\frac{1}{\sigma}|j\widetilde{\sigma} - \mu| \leq \frac{\widetilde{\sigma}}{2\sigma} \implies \mu \in [(j - 0.5)\widetilde{\sigma}, (j + 0.5)\widetilde{\sigma}].$$

$\square$

**Proposition D.6.** *Fix some univariate Gaussian $g = \mathcal{N}(0, \sigma^2)$. Partition $\mathbb{R}_{>0}$ into disjoint bins $\{B_i\}_{i\in\mathbb{Z}}$ where $B_i = (2^i, 2^{i+1}]$ and let $j \in \mathbb{N}$ satisfy $2^j < \sigma \leq 2^{j+1}$. It follows that:*

$$\mathbf{P}_{X\sim g}[|X| \in B_j] \geq \frac{1}{4}.$$

*Proof.* Since $2^j < \sigma \leq 2^{j+1}$, we can write $\sigma = 2^{j+c}$ for some $c \in (0, 1]$. Let $x = 2^{-c}$ and notice $x \in [1/2, 1)$. We have the following:

$$\mathbf{P}_{X\sim g}[|X| \in B_j] = 2\left(\Phi\left(\frac{2^{j+1}}{\sigma}\right) - \Phi\left(\frac{2^j}{\sigma}\right)\right)$$

$$= 2\left(\Phi\left(2^{1-c}\right) - \Phi\left(2^{-c}\right)\right)$$

$$= 2f(2^{-c}), \tag{3}$$

where we define $f(x) = \Phi(2x) - \Phi(x)$. We now aim to lower bound $f(x)$. By taking the derivative of $f(x)$ twice, we have that $f''(x) = \sqrt{(1/2\pi)}(x\exp(-x^2/2) - 8x\exp(-2x^2))$. By a simple calculation, we have that $f''(x) \leq 0$ when $x \in [0, 2\ln 8/3] \supset [1/2, 1)$, so $f(x)$ is concave when $x \in [1/2, 1)$. This implies that $f(x) \geq \min\{f(1/2), f(1)\}$ for any $x \in [1/2, 1)$, so from Eq. (3) we have

$$\mathbf{P}_{X\sim g}[|X| \in B_j] \geq 2\min\{f(1/2), f(1)\}$$

$$= 2\min\left\{\Phi(1) - \Phi\left(\frac{1}{2}\right), \Phi(2) - \Phi(1)\right\}$$

$$> \frac{1}{4},$$

where the last inequality follows from a direct calculation. $\square$

**Proposition D.7.** *Fix $g = \mathcal{N}(\mu, \sigma^2)$ and $g' \in \mathcal{H}_\gamma(g)$. Let $Z = (X_1 - X_2)/\sqrt{2}$ where $X_1, X_2 \sim g'$ i.i.d. Let $Y \sim \mathcal{N}(0, \sigma^2)$. Then for any measurable $S \subseteq \mathbb{R}$*

$$\mathbf{P}[|Z| \in S] \geq (1 - \gamma)^2 \cdot \mathbf{P}[|Y| \in S].$$

*Proof.* We prove this via a coupling argument. Since $g' \in \mathcal{H}_\gamma(g)$ we have $g' = (1 - \gamma)g + \gamma h$ for some distribution $h$.

Let $Y_1, Y_2 \sim g$ i.i.d. so that $Y = \frac{Y_1 - Y_2}{\sqrt{2}} \sim \mathcal{N}(0, \sigma^2)$. Also, let $H_1, H_2 \sim h$ i.i.d. Finally, let $B_1, B_2$ be independent Bernoulli random variables with parameter $1 - \gamma$, i.e. $B_i = 1$ with probability $1 - \gamma$ and $B_i = 0$ with probability $\gamma$.

Now let $X_i = Y_i \cdot B_i + H_i \cdot (1 - B_i)$ and note that $X_i \sim g'$. If $B_1 = B_2 = 1$ and $|Y| \in S$ then certainly $|Z| = |X_1 - X_2|/\sqrt{2} \in S$. Hence,

$$\mathbf{P}[|Z| \in S] \geq \mathbf{P}[\{B_1 = 1\} \cap \{B_2 = 1\} \cap \{|Y| \in S\}] = (1 - \gamma)^2 \mathbf{P}[|Y| \in S],$$

where the last equality uses the fact that $B_1, B_2, Y$ are mutually independent random variables. $\qquad\square$

# E  Omitted Results from Section 5

---

**Algorithm 5:** `Multivariate-Gaussian-Decoder`$(\alpha, \beta, \gamma, \varepsilon, \delta, D)$.

---

**Input**  : Parameters $\varepsilon, \alpha, \beta, \gamma \in (0, 1), \delta \in (0, 1/n)$, and a dataset $D$

**Output** : Set of distributions $\widehat{\mathcal{F}} \subset \mathcal{G}^d$.

1 Initialize $\widehat{V}_j \leftarrow \emptyset$, $\widehat{M}_j \leftarrow \emptyset$ for $j \in [d]$

2 Set $D_i \leftarrow \{X_i : X \in D\}$ for $i \in [d]$             // Split dataset by dimension.

3 For $i \in [d]$ **do**

4     $\widehat{M}_i, \widehat{V}_i \leftarrow$ `Univariate-Gaussian-Decoder`$(\alpha/d, \beta/d, \gamma, \varepsilon/d, \delta/d, D_i)$

5 $\widehat{M} \leftarrow \{(\widehat{\mu}_1, \ldots, \widehat{\mu}_d) : \widehat{\mu}_i \in \widehat{M}_i, i \in [d]\}$

6 $\widehat{\Lambda} \leftarrow \{\text{diag}(\widehat{\sigma}_1^2, \ldots, \widehat{\sigma}_d^2) : \widehat{\sigma}_i \in \widehat{V}_i, i \in [d]\}$

7 $\widehat{\mathcal{F}} \leftarrow \left\{\mathcal{N}(\widehat{\mu}, \widehat{\Sigma}) : \widehat{\mu} \in \widehat{M}, \widehat{\Sigma} \in \widehat{\Lambda}\right\}$

8 **Return** $\widehat{\mathcal{F}}$

---

*Proof of Lemma 5.2.* The list-decoding algorithm for multivariate Gaussians is given by Algorithm 5.

**Privacy.**  We first prove the algorithm is $(\varepsilon, \delta)$-DP. By the guarantee of Lemma 4.8, each run of line 4 in the loop is $(\varepsilon/d, \delta/d)$-DP. No subsequent part of the algorithm accesses the data, so by post processing (Lemma 2.9) and basic composition (Lemma 2.8) the entire algorithm is $(\varepsilon, \delta)$-DP.

**Bound on $|\widehat{\mathcal{F}}|$.**  We now prove the claimed upper bound on the size of $\widehat{\mathcal{F}}$. By the guarantee of Lemma 4.8, each $\widehat{M}_i$ and $\widehat{V}_i$ obtained on line 4 satisfy $|\widehat{M}_i| \leq 144 \cdot (2 \cdot \lceil d/\alpha \rceil + 1)/(1 - \gamma)^3$ and $|\widehat{V}_i| \leq 12 \cdot \lceil \log_{1+\alpha/d}(2) \rceil /(1 - \gamma)^2$. This immediately gives us

$$|\widetilde{\mathcal{F}}| = |\widehat{M}| \cdot |\widehat{\Lambda}| = \left(\prod_{i=1}^{d} |\widehat{M}_i|\right) \cdot \left(\prod_{i=1}^{d} |\widehat{V}_i|\right) \leq \left(\left(\frac{1728}{(1-\gamma)^5}\right) \cdot \left\lceil \log_{1+\alpha/d}(2) \right\rceil \cdot (2 \cdot \lceil d/\alpha \rceil + 1)\right)^d.$$

To get the bound on $L = |\widehat{\mathcal{F}}|$ as stated in the lemma, we use the fact that $\log_{1+\alpha/d}(2) = \frac{\ln(2)}{\ln(1+\alpha/d)} \leq \frac{2\ln(2)}{\alpha/d}$, where the inequality uses the fact that $\ln(1 + x) \geq x/2$ for $x \in [0, 1]$.

**Utility and sample complexity.**  We now prove that the algorithm is a list-decodable learner. Fix some $g = \prod_{i=1}^{d} \mathcal{N}(\mu_i, \sigma_i^2) \in \mathcal{G}^d$ and $g' \in \mathcal{H}_\gamma(g)$. By our choice of parameters and the guarantee of Lemma 4.8, a single run of algorithm `Univariate-Gaussian-Decoder` on line 4 outputs lists $\widehat{M}_i$ and $\widehat{V}_i$ such that there exist $\widehat{\mu}_i \in \widehat{M}_i$ and $\widehat{\sigma}_i \in \widehat{V}_i$ satisfying $|\widehat{\mu}_i - \mu_i| \leq \alpha\sigma_i/d$ and $|\widehat{\sigma}_i - \sigma_i| \leq \alpha\sigma_i/d$ with probability at least $1 - \beta/d$ so long as

$$n = \Omega\left(\frac{d\log(d/\beta\delta)}{(1-\gamma)^2\varepsilon} + \frac{d\log(d/(1-\gamma)\beta\delta)\sqrt{\log(d/(1-\gamma)\delta)}}{(1-\gamma)^2\varepsilon}\right).$$

By a union bound, we have with probability no less than $1 - \beta$ that for all $i \in [d]$, $|\widehat{\mu}_i - \mu_i| \leq \alpha\sigma_i/d$ and $|\widehat{\sigma}_i - \sigma_i| \leq \alpha\sigma_i/d$. By a standard argument, this implies that with probability at least $1 - \beta$ there is some $\widehat{g} \in \widehat{\mathcal{F}}$ such that $d_{\text{TV}}(\widehat{g}, g) \leq \alpha$ (see Proposition A.1 and Proposition A.2). $\qquad\square$

# F  Learning Mixtures of Gaussians with Known Covariance

In this section, we prove the following result, which is a formal version of Theorem 1.2. Let $\mathcal{G}_1^d$ be the class of Gaussians with identity covariance matrix.

**Theorem F.1.** *For any $\varepsilon \in (0,1)$ and $\delta \in (0, 1/n)$, there is an $(\varepsilon, \delta)$-DP PAC learner for $k$-mix$\left(\mathcal{G}_1^d\right)$ that uses*

$$m(\alpha, \beta, \varepsilon, \delta) = \widetilde{O}\left(\frac{kd\log(1/\beta)}{\alpha^2} + \frac{kd + \log(1/\beta\delta)}{\alpha\varepsilon}\right)$$

*samples.*

Note that the theorem also implies the case where the covariance matrix $\Sigma$ is an arbitrary but known covariance matrix. Indeed, given samples $X_1, \ldots, X_m$, one can apply the algorithm of Theorem F.1 to $\Sigma^{-1/2}X_1, \ldots, \Sigma^{-1/2}X_m$ instead.

The proof of Theorem F.1 follows from Theorem 3.1 and Corollary F.2, which is a corollary of Lemma 4.2.

**Corollary F.2.** *For any $\varepsilon \in (0,1)$ and $\delta \in (0, 1/n)$, there is an $(\varepsilon, \delta)$-DP L-list-decodable learner for $\mathcal{G}_1^d$ where $L = O(d/(1-\gamma)\alpha)^d$, and the number of samples used is*

$$m_{\mathrm{LIST}}(\alpha, \beta, \gamma, \varepsilon, \delta) = O\left(\frac{d\log(d/\beta\delta)}{(1-\gamma)\varepsilon}\right).$$

*Proof.* For each $i \in [d]$ let $D_i = \{X_i : X \in D\}$ be the dataset consisting of the $i$th coordinate of each element in $D$. We run Univariate-Mean-Decoder$(\varepsilon/d, \delta/d, \beta/d, \gamma, \sigma, D_i)$ to obtain the set $\widetilde{M}_i$. Let $\widehat{M}_i$ be an $\alpha/d$-net of the set of intervals $\{[\widetilde{\mu}_i - 1, \widetilde{\mu}_i + 1] : \widetilde{\mu}_i \in \widetilde{M}_i\}$ of size $|\widetilde{M}_i| \cdot (2 \cdot \lceil d/2\alpha \rceil + 1)$, i.e.

$$\widehat{M}_i = \{\widetilde{\mu}_i + 2j\alpha/d : \widetilde{\mu}_i \in \widetilde{M}_i, \ j \in \{0, \pm 1, \ldots, \pm\lceil d/2\alpha \rceil\}\}.$$

Let $\widehat{M} = \{(\widehat{\mu}_1, \ldots, \widehat{\mu}_d) : \widehat{\mu}_i \in \widehat{M}_i\}$. We then return $\widehat{\mathcal{F}} = \{\mathcal{N}(\widehat{\mu}, I) : \widehat{\mu} \in \widehat{M}\}$. Finally, Lemma 4.2 (with a union bound over the $d$ coordinates), basic composition (Lemma 2.8), and post-processing (Lemma 2.9) imply that the algorithm is $(\varepsilon, \delta)$-DP while Lemma 4.2, Proposition A.2, and Proposition A.1 imply the accuracy guarantee. □