# OpenReview forum: "Privately Learning Mixtures of Axis-Aligned Gaussians"
_NeurIPS.cc/2021/Conference — NeurIPS 2021 Poster_

### Official Review · Reviewer_Xmc7 · 2021-06-30

**Rating:** 7
**Confidence:** 4

**Summary:**

As the title suggests, this paper gives algorithms for learning mixtures of $k$ Gaussians to within total variation distance $\alpha$. It uses a reduction to private list decodable learners from privately learning mixtures of distributions. The number of samples used is linear in $d$ and quadratic in $k$. The flow of the paper is as follows: first it gives algorithms for estimating mixtures of univariate Gaussians, then uses them to estimate mixtures of axis-aligned Gaussians. The main, non-trivial prior tool used in this paper is the private hypothesis selection algorithm. The algorithm for estimating mixtures of univariate Gaussians first involves creating a list-decodable learner for univariate Gaussians, followed by using the aforementioned reduction.

**Limitations And Societal Impact:**

This is a theoretical paper, therefore, it's hard to say what the negative impacts on the society would be. In my opinion, there shouldn't be any, and the authors claim the same themselves.

**Main Review:**

Update: Based on the authors' response, I have decided to update my final evaluation.

Strengths:
1. The paper is very theoretical, and has non-trivial ideas for both proofs and algorithms.
2. The sample complexity achieved is very close to the non-private, and is an improvement over Kamath et al.'s paper on learning mixtures of Gaussians.
3. The reduction is very useful and general, and could be used for learning mixtures of other kinds of distributions as well.
4. The paper, in general, is quite well written. I was able to understand everything in one shot. Section 3 is very helpful in understanding the rest of the paper.
5. It was important to mention that the class of mixtures of univariate Gaussians does not permit locally small covers. This helped me understand why it was important to look into this problem more closely.

Weaknesses:
1. Although the sample complexity is quite good, the algorithms run possibly in exponential time. hence, they are not practical.
2. Some proof ideas for the main theorems (like Theorem 4.6) could have been put in, as well.
3. A nitpick comment. When I usually see exponential time algorithms in differential privacy literature, they are usually optimal or close to being optimal. Here, in terms of $k$ and $\alpha$, the private sample complexity seems sub-optimal. That could be improved. However, it is a big improvement over [KSSU'19].
4. Another nitpick comment. Generalisation to arbitrary mixing weights could have been mentioned, too, for the sake of completeness, although I'm aware of the space restrictions.

**Time Spent Reviewing:**

4

---

> ### Author Response · Authors · 2021-08-10
> **Response to Reviewer Xmc7**
>
> We would like to thank the reviewer for their careful and detailed review. In regards to achieving the conjectured optimal sample complexity, this seems to be challenging to achieve with our approach, and we explained this in great detail in our response to reviewer XGxw. We welcome reviewer Xmc7 to take a look and update their opinion based on these comments. Finally, we will also try to add some of the details regarding generalizing to arbitrary mixing weights to the paper, subject to space requirements.

---

### Official Review · Reviewer_GCJ5 · 2021-07-18

**Rating:** 7
**Confidence:** 3

**Summary:**

This paper studies differentially-private learning of mixture models -- mixtures of Gaussians, in particular, but many ideas likely extend beyond that setting.
The paper introduces new ideas to DP learning of mixture models when the mixture components have a priori unbounded means and (co)variances.



**Limitations And Societal Impact:**

The technical ideas are nice but not super surprising or hard.
Still, I think the conceptual contribution is clear enough that this should not keep the paper from being accepted.

**Main Review:**

This paper studies differentially-private learning of mixture models -- mixtures of Gaussians, in particular, but many ideas likely extend beyond that setting.
Mixture models are the canonical theoretical setting for studying non-homogeneous data sets, and differential privacy is the industry-standard mathematically rigorous notion of privacy, so this is a learning problem of fundamental importance.

The context for the paper is the following state of affairs in private mixture-model learning.
First of all, learning mixtures of Gaussians under the stringent notion of *pure* DP is known to require *a priori* boundedness of the means and covariances.
This paper studies private learning without such a prior bounds, which means that one must relax pure DP to the weaker $(\epsilon,\delta)$-DP.
(Allowing additive in addition to multiplicative privacy loss.)

In the pure DP setting, the canonical approach for learning distributions (at least without worrying about running times) involves finite covers: this technology readily generalizes from learning a class of distributions to learning the class of mixtures of those distributions.
But this is not true in the $(\epsilon,\delta)$-DP setting: the "locally small covers" approach which works for learning (potentially unbounded) Gaussians does not generalize to Gaussian mixtures.
So a new idea is needed to show that (potentially unbounded) Gaussian mixtures can be learned without too many samples.

This paper introduces a new approach to show that this holds at least for univariate and high-dimensional but axis-aligned Gaussian mixtures.
The approach is conceptually nice.
The authors use the (well known) observation that learning mixture models can be reduced to a harder problem, "list-decodable learning", where the goal is to learn a distribution given a list of samples of which a large majority have been corrupted by a malicious adversary.
Since this is not possible information-theoretically, instead the goal is just to learn a short list of hypotheses of which one must be close to the true underlying distribution.

The authors observe that (1) the known reduction from mixture model learning to list-decodable learning can be done privately, using private hypothesis selection, and (2) design a differentially private algorithm for list-decodable learning of (potentially unbounded) Gaussians.
Most of the work for (2) is in the univariate case; algorithms for learning univariate Gaussians often generalize readily to high-dimensional axis-aligned Gaussians and that is the case here.

This paper provides nice technical progress on a fundamental problem -- it was clear that existing cover-based methods would not work, and this paper provides a new idea.
I think it should be accepted.

**Time Spent Reviewing:**

2 hours

---

> ### Author Response · Authors · 2021-08-10
> **Response to Reviewer GCJ5**
>
> We would like to thank the reviewer for their careful and detailed review. We very much appreciate the positive comments and feedback.

---

### Official Review · Reviewer_XGxw · 2021-07-18

**Rating:** 6
**Confidence:** 4

**Summary:**

This paper considered the problem of learning a mixture of gaussians under DP constraint. The authors proposed an (exponential time) algorithm for learning univariate k-mixture of gaussian distributions with sample complexity k^2log^(3/2)(1/\delta)/alpha^2eps. They conjectured the extra k factor in the sample complexity is information theoretically unnecessary. The key of the algorithm is a reduction from the list-decodable learning to learning mixture distributions. Specifically, given a list decodable algorithm, one can enumerate all possible combinations of the k-mixtures and use a private hypothesis selection algorithm to find the most plausible k-mixture. They proposed a list decodable algorithm for learning gaussian distribution, which adopts a similar approach as [48].

For the multivariate gaussian setting, they considered the axis-aligned gaussian setting. They first obtain a list of distribution on each dimension, and take all possible combinations of the univariate distribution to form a large list of axis aligned gaussians. This leads to a multivariate gaussian k-mixture learning algorithm with sample complexity k^2dlog^(3/2)(1/delta)/alpha^2eps.

**Limitations And Societal Impact:**

Yes

**Main Review:**

The paper provides new results on an interesting and fundamental problem. It is a little bit disappointing that very few new ideas are applied in obtaining the result except the connection to list decodable learning which has been similarly observed before. It is also a bit unsatisfactory that the algorithm, even though taking exponential time, is not able to get the optimal sample complexity. I am curious about the authors' thoughts on extending the approach to remove the extra k factor and general (non axis-aligned) mixed gaussian setting. For now I will vote for a weak accept.


**Time Spent Reviewing:**

3

---

> ### Author Response · Authors · 2021-08-10
> **Response to Reviewer XGxw**
>
> We thank the reviewer for carefully reviewing our manuscript and the thoughtful review. The dependence on $k$ is something that we thought a lot about and it seems to be quite technically challenging to obtain a subquadratic dependence on $k$. One bottleneck seems to be the problem of (privately) outputting a list of candidate variances, even for uniform mixtures of Gaussians, i.e. weight of $1/k$ on each component. The natural way, which we do in our paper, is to sample two points $X_1$, $X_2$, and use $(X_{1}-X_{2})/\sqrt{2}$ as a potential estimate. But this is only a good estimate of the standard deviation with probability $1/k^2$ (both $X_1$, $X_2$ need to come from the same component).
> Another approach may be to draw $X_{1}, \dots, X_{m}$ and then look at $(X_{i} - X_{j}) / \sqrt{2}$ for all pairs $i,j$. But it turns out this approach requires m on the order of $k^2$ so once again, we are stuck with a quadratic dependence.
>
> There is actually a second bottleneck that is preventing us from obtaining the optimal linear dependence on $k$. Suppose that the variances are known beforehand but, importantly, the variances may all be distinct. With the techniques in our paper (and with advanced composition), we are only able to obtain a $k^{3/2}$ dependence on $k$. So even if we find an efficient  list-decodable learner for the variances, there are still difficulties for learning the mean when the variances are different.
>
> With regards to high-dimensional covariance matrices: the problem is significantly more complicated. In particular, with our histogram approach, there is the question of discretization. In one dimension, discretizing is fairly straightforward. In higher-dimensions, it is not clear because one needs to capture both the eigenvalues of the covariance matrix as well as the eigenvectors. (In one dimension, the eigenvectors are trivial.)
>
> In fact, even for a single d-dimensional Gaussian with arbitrary covariance matrix, we only know of one method to privately estimate the covariance matrix with a sample complexity that does not require a priori bounds on the condition number, and this approach is based on a local covering technique [AAK21]. It is not clear whether or not this can be ported to this setting because the local covering approach provides a very large (and possibly infinite!) list of candidate distributions which is at odds with the requirements of the list-decodable setting.
>
> So we believe both questions that the reviewer has are very interesting and would be very interesting for future work.
>
> [AAK21] Aden-Ali, Ishaq, Hassan Ashtiani, and Gautam Kamath. "On the sample complexity of privately learning unbounded high-dimensional Gaussians." Algorithmic Learning Theory. PMLR, 2021.

---

### Decision · Program_Chairs · 2021-09-27

**Decision:**

Accept (Poster)

**Comment:**

This paper makes progress on a fundamental problem: Learning mixtures of axis aligned Gaussian distributions under (eps, delta) differential privacy. The paper is a nice combination of two ideas: reduction to list decodable learning, and a private algorithm for list decodable learning. These ideas (particularly the second) should find applications in other places. It would be nice if the authors add a comment about the optimal sample complexity (the extra k factor).